# MAIT cells regulate NK cell-mediated tumor immunity

Emma V. Petley[1,2], Hui-Fern Koay [3,4], Melissa A. Henderson[1,2], Kevin Sek [1,2], Kirsten L. Todd[1,2], Simon P. Keam[1,2,5], Junyun Lai[1,2], Imran G. House [1,2], Jasmine Li[1,2], Magnus Zethoven[2,6], Amanda X. Y. Chen[1,2], Amanda J. Oliver[1,2], Jessica Michie[1,2], Andrew J. Freeman [1,2], Lauren Giuffrida[1,2], Jack D. Chan [1,2], Angela Pizzolla [1,2], Jeffrey Y. W. Mak [7,8], Timothy R. McCulloch [9], Fernando Souza-Fonseca-Guimaraes[9], Conor J. Kearney[1,2], Rosemary Millen[1,2], Robert G. Ramsay[1,2], Nicholas D. Huntington [10,11,12], James McCluskey [3], Jane Oliaro [1,2,13], David P. Fairlie [7,8], Paul J. Neeson [1,2], Dale I. Godfrey [3,4,15✉], Paul A. Beavis [1,2,14,15✉] & Phillip K. Darcy [1,2,12,14,15✉]

The function of MR1-restricted mucosal-associated invariant T (MAIT) cells in tumor immunity is unclear. Here we show that MAIT cell-deficient mice have enhanced NK cell-dependent control of metastatic B16F10 tumor growth relative to control mice. Analyses of this interplay in human tumor samples reveal that high expression of a MAIT cell gene signature negatively impacts the prognostic significance of NK cells. Paradoxically, pre-pulsing tumors with MAIT cell antigens, or activating MAIT cells in vivo, enhances anti-tumor immunity in B16F10 and E0771 mouse tumor models, including in the context of established metastasis. These effects are associated with enhanced NK cell responses and increased expression of both IFN-γ-dependent and inflammatory genes in NK cells. Importantly, activated human MAIT cells also promote the function of NK cells isolated from patient tumor samples. Our results thus describe an activation-dependent, MAIT cell-mediated regulation of NK cells, and suggest a potential therapeutic avenue for cancer treatment.

[1] Cancer Immunology Program, Peter MacCallum Cancer Centre, Melbourne, VIC, Australia. [2] Sir Peter MacCallum Department of Oncology, The University of Melbourne, Parkville, VIC, Australia. [3] Department of Microbiology & Immunology, Peter Doherty Institute for Infection and Immunity, University of Melbourne, Melbourne, VIC, Australia. [4] Australian Research Council Centre of Excellence in Advanced Molecular Imaging, University of Melbourne, Melbourne, VIC, Australia. [5] Tumour Suppression and Cancer Sex Disparity Laboratory, Peter MacCallum Cancer Centre, Melbourne, VIC, Australia. [6] Bioinformatics Core Facility, Peter MacCallum Cancer Centre, Melbourne, VIC, Australia. [7] Institute for Molecular Bioscience, The University of Queensland, Brisbane, QLD, Australia. [8] Australian Research Council Centre of Excellence in Advanced Molecular Imaging, The University of Queensland, Brisbane, QLD, Australia. [9] University of Queensland Diamantina Institute, The University of Queensland, Brisbane, QLD, Australia. [10] Department of Medical Biology, Faculty of Medicine, Dentistry and Health Sciences, University of Melbourne, Melbourne, VIC, Australia. [11] Division of Molecular Immunology, Walter and Eliza Hall Institute of Medical Research, Melbourne, VIC, Australia. [12] Biomedicine Discovery Institute and the Department of Biochemistry and Molecular Biology, Monash University, Melbourne, VIC, Australia. [13] Department of Immunology, Monash University, Melbourne, VIC, Australia. [14] Department of Pathology, University of Melbourne, Melbourne, VIC, Australia. [15] These authors contributed equally: Dale I. Godfrey, Paul A. Beavis, Phillip K. Darcy. ✉email: godfrey@unimelb.edu.au; paul.beavis@petermac.org; phil.darcy@petermac.org

Mucosal-associated invariant T (MAIT) cells are unconventional T cells that express a semi-invariant T-cell receptor (TCR) α-chain (Vα19-Jα33 in mice, Vα7.2-Jα33/Jα12/Jα20 in humans) paired with a limited TCR β-chain repertoire[1–4]. This TCR specificity restricts MAIT cells to antigens presented by the non-polymorphic MHC class-I related (MR1) molecule, which is highly conserved across mammals[5–8]. The pathogenic and protective roles of MAIT cells in autoimmune and inflammatory disease have been extensively studied[9–12] but their role in cancer is unclear. Although there are currently no reports of tumor antigens presented by MR1 to MAIT cells[13–16], MAIT cells upregulate granzyme B and perforin following activation[5,17,18] and have the capacity to kill tumor cells pulsed with the MAIT cell antigen 5-(2-oxopropylideneamino)-6-D-ribitylaminouracil (5-OP-RU)[19], suggesting that MAIT cells are capable of eliciting direct anti-tumor immune responses. Furthermore, MAIT cells can be activated by inflammatory cytokines, such as IL-12, IL-18, and TL1A[20,21], and express receptors known to activate NK cells following interaction with their respective ligands expressed on tumor cells, including DNAM-1[22,23]. These observations suggest that MAIT cells may be activated within the tumor microenvironment (TME) through alternative mechanisms to direct recognition of tumor antigens and have the potential to detect tumor cells other than through presentation of antigens by MR1. Upon activation, MAIT cells have the potential to secrete cytokines predominantly associated with both anti-tumor (e.g., IFN-γ and TNF) and pro-tumor activity (e.g., IL-17α, IL-22, and IL-13)[6,22,24]. Therefore, although it is plausible that MAIT cells may become activated within the TME, their contribution to anti-tumor immune responses remains unclear due to this diverse range of effector functions.

Human MAIT cells, characterized by expression of Vα7.2 and CD161[25], have been found within primary and metastatic lesions in a range of tumor types. However, direct evidence of their role in cancer is limited[9]. High infiltration of CD8+CD161+ T cells is associated with good prognosis across a large range of diverse tumor samples, suggesting tumor-infiltrating MAIT cells may correlate with favorable outcomes[26]. Numbers of circulating MAIT cells are significantly reduced in patients with mucosal-associated cancers, including colon and lung, and correlated with increased MAIT cell infiltration in tumors[27–29]. Interestingly, MAIT cells isolated from colorectal cancer (CRC) lesions and liver metastases expressed lower IFN-γ levels compared to MAIT cells from healthy tissue[27,29,30], suggesting that their function may be affected by the immunosuppressive TME. Furthermore, CRC patients with high MAIT cell tumor infiltration have a less favorable clinical outcome compared to patients with a lower MAIT cell infiltration[31]. Conversely, a study in patients with hepatocellular carcinoma suggests that increased MAIT cell infiltration correlated with an improved prognosis[32]. Such divergent observations have been observed across numerous studies[9], making it difficult to know whether MAIT cells play a positive or negative role in anti-tumor immunity. This highlights the need to investigate the functional role of MAIT cells in mouse tumor models where they can be directly manipulated.

In this study, we investigate the function of MAIT cells in murine tumor models, where we reveal a dichotomous role for MAIT cells in anti-tumor immunity. MAIT cell-deficient (B6-MAIT$_{cast}$ MR1$^{-/-}$) mice exhibit enhanced resistance to tumor growth, suggesting an inhibitory role for MAIT cells in tumor immunity. Conversely, MAIT cell activation and expansion via exposure to free 5-OP-RU antigen or 5-OP-RU-pulsed tumor cells enhances anti-tumor immunity. In both cases, these effects are mediated by modulation of NK cell activity in an IFN-γ-dependent manner. Moreover, we provide evidence for a similar function of MAIT cells in human cancer. Devising novel

strategies to enhance MAIT cell activation represents a potential therapy for cancer, particularly those which originate in, or metastasize to, tissues with high levels of MAIT cells, such as the liver and lung.

## Results

**MAIT cell-deficient mice are protected against tumor growth.** To assess the potential role of MAIT cells in controlling anti-tumor immunity, we utilized the congenic B6-MAIT$_{cast}$ MR1 WT mouse strain, which are reported to have increased numbers of MAIT cells relative to wildtype C57BL/6 mice and thus the numbers of MAIT cells in these mice are more similar to those observed in humans[33]. We utilized these mouse strains to examine the potential role of MAIT cells in modulating lung tumor metastasis. B6-MAIT$_{cast}$ MR1$^{-/-}$ mice, which lack MAIT cells, were significantly protected in an experimental metastasis model involving intravenous inoculation of B16F10 melanoma cells, relative to B6-MAIT$_{cast}$ MR1 WT mice (Fig. 1a). In a distinct model involving subcutaneous growth of low dose B16F10 melanoma cells, we again found that tumor growth was significantly inhibited in B6-MAIT$_{cast}$ MR1$^{-/-}$ mice relative to B6-MAIT$_{cast}$ MR1 WT mice, leading to enhanced survival of MAIT cell-deficient mice (Fig. 1b). These findings suggest that MAIT cells play a pro-tumor role in both a lung metastasis and subcutaneous model. Given these effects, we compared the immune landscape in the lungs of B6-MAIT$_{cast}$ MR1 WT and B6-MAIT$_{cast}$ MR1$^{-/-}$ mice. As expected, we confirmed that B6-MAIT$_{cast}$ MR1 WT mice had significantly increased numbers of MAIT cells relative to C57BL/6 mice and B6-MAIT$_{cast}$ MR1$^{-/-}$ mice (Fig. 1c, d and Supplementary Fig. 1a). We observed a threefold increase in MAIT cell numbers between B6-MAIT$_{cast}$ MR1 WT mice and C57BL/6 mice as evaluated by MR1-5-OP-RU tetramer staining, consistent with a previous report, which estimated MAIT cell frequency based upon Vα19-Jα33 mRNA expression in these mice strains[33]. Comparison of other immune subsets in the lungs of B6-MAIT$_{cast}$ MR1 WT and B6-MAIT$_{cast}$ MR1$^{-/-}$ mice revealed no differences in the numbers of γδ T cells, NKT cells, CD11b+Ly6C+ myeloid cells and both CD4+ and CD8+ conventional αβ T cells (Tconv) (Fig. 1c, d). However, there was a significant increase in numbers of NK cells in B6-MAIT$_{cast}$ MR1$^{-/-}$ mice relative to B6-MAIT$_{cast}$ MR1 WT mice (Fig. 1e). Further analysis of this NK cell population revealed that B6-MAIT$_{cast}$ MR1$^{-/-}$ mice exhibited an increased proportion and absolute number of NK cells exhibiting the CD11b+CD27− phenotype that marks an NK cell subset with enhanced cytotoxic function (Fig. 1e)[34,35]. However, the expression of NK cell receptors DNAM-1, CD96 and TIGIT, which contribute to NK cell-mediated anti-tumor function[36], was similar on NK cells from B6-MAIT$_{cast}$ MR1 WT and B6-MAIT$_{cast}$ MR1$^{-/-}$ mice, while a slight decrease in NKG2D expression on NK cells from B6-MAIT$_{cast}$ MR1$^{-/-}$ mice was observed (Supplementary Fig. 1b). This suggests that the lack of MAIT cells predominantly affected NK cell numbers and maturation status. Using a more comprehensive gating strategy to discriminate NK1.1+ conventional NK cells from ILC1s, we confirmed that these NK1.1+ cells expressed Eomes, confirming their identity as NK cells (Supplementary Fig. 1c). Interestingly, this analysis also revealed that MAIT cells isolated from the lung expressed DNAM-1 and CD96 (Supplementary Fig. 1d). Given this NK cell phenotype, we investigated whether NK cells were responsible for the protected phenotype of B6-MAIT$_{cast}$ MR1$^{-/-}$ mice. We therefore assessed B16F10 tumor metastasis in B6-MAIT$_{cast}$ MR1 WT and B6-MAIT$_{cast}$ MR1$^{-/-}$ mice in the context of NK cell depletion using anti-asialo GM-1 (αASGM1), after confirming that αASGM1 did not deplete MAIT cells (Supplementary Fig. 1e). In this setting,

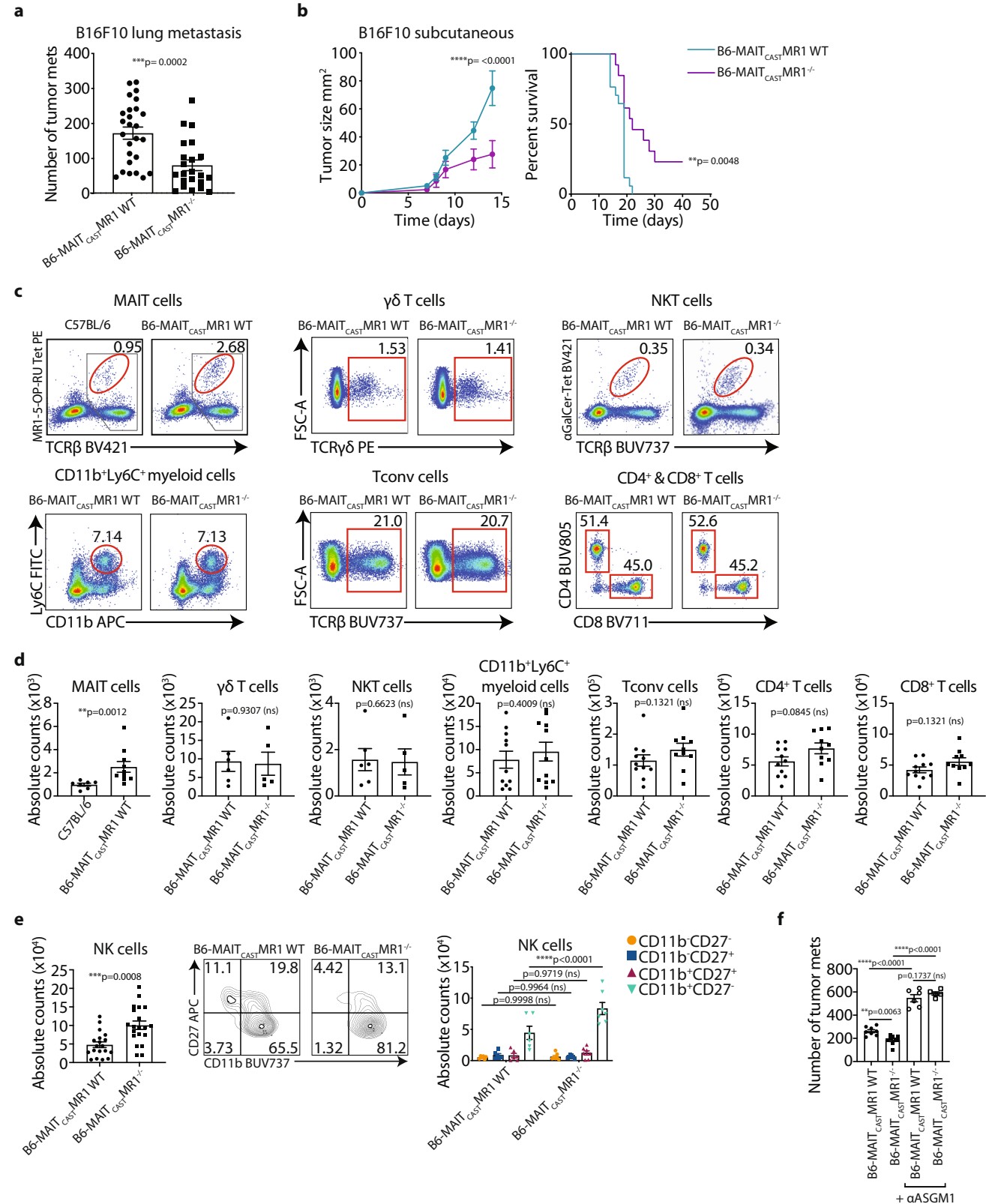

the number of metastases were markedly increased following NK cell depletion as expected based upon previous observations[37]. Furthermore, following NK cell depletion B6-MAIT$_{cast}$ MR1$^{-/-}$ and B6-MAIT$_{cast}$ MR1 WT mice exhibited a similar metastatic burden following B16F10 tumor challenge, indicating that the enhanced protection exhibited by B6-MAIT$_{cast}$ MR1$^{-/-}$ mice is

NK cell dependent (Fig. 1f and Supplementary Fig. 1f). Similarly, using the low dose of B16F10 in the subcutaneous model we also observed NK cell-dependent anti-tumor immunity (Supplementary Fig. 1g). To investigate the mechanism by which B6-MAIT$_{cast}$ MR1$^{-/-}$ mice were protected against B16F10 lung metastasis, we investigated the phenotype of NK cells in tumor-

**Fig. 1 MAIT cell-deficient mice elicit enhanced NK cell-mediated anti-tumor immune responses. a** $2 \times 10^5$ B16F10 tumor cells were injected i.v. into B6-MAIT$_{cast}$ MR1 WT or B6-MAIT$_{cast}$ MR1$^{-/-}$ mice. Lungs were harvested and the number of tumor metastases enumerated 14 days after inoculation. Data is presented as mean ± SEM of $n = 26$ (B6-MAIT$_{cast}$ MR1 WT) or $n = 22$ (B6-MAIT$_{cast}$ MR1$^{-/-}$) independent mice from four combined experiments, two-tailed Mann–Whitney test. **b** $3 \times 10^4$ B16F10 tumor cells were injected s.c. into B6-MAIT$_{cast}$ MR1 WT or B6-MAIT$_{cast}$ MR1$^{-/-}$ mice and tumor size monitored. Data is presented as mean ± SEM of $n = 10$ (B6-MAIT$_{cast}$ MR1 WT) or $n = 6$ (B6-MAIT$_{cast}$ MR1$^{-/-}$) independent mice from a representative experiment of $n = 2$, Two-way ANOVA. Survival was defined when tumor size exceeded 100 mm$^2$. Data is presented as mean ± SEM of $n = 17$ (B6-MAIT$_{cast}$ MR1 WT) or $n = 13$ (B6-MAIT$_{cast}$ MR1$^{-/-}$) independent mice from two combined experiments, Log-rank (Mantel-Cox) test. **c, d** Representative flow cytometry plots (**c**) and absolute numbers (**d**) of MAIT (CD45.2$^+$CD19$^-$TCRβ$^+$MR1-5-OP-RU-tetramer$^+$), γδ T (CD45.2$^+$CD19$^-$TCRγδ$^+$), NKT (CD45.2$^+$CD19$^-$TCRβ$^+$αGalCer-tetramer$^+$), myeloid (CD45.2$^+$CD19$^-$CD11b$^+$Ly6C$^+$), conventional T (Tconv; CD45.2$^+$CD19$^-$TCRβ$^+$) and CD4$^+$ and CD8$^+$ T cells in the lungs of indicated mice. Data is presented as mean ± SEM of $n = 7$ (C57BL/6) and $n = 10$ (B6-MAIT$_{cast}$ MR1 WT) independent mice from three combined experiments (MAIT cells), or $n = 6$ (B6-MAIT$_{cast}$ MR1 WT) and $n = 5$ (B6-MAIT$_{cast}$ MR1$^{-/-}$) independent mice from two combined experiments (γδ T cells, NKT cells), or $n = 11$ independent mice from three combined experiments (CD11b$^+$Ly6C$^+$ myeloid cells), or $n = 11$ (B6-MAIT$_{cast}$ MR1 WT) and $n = 10$ (B6-MAIT$_{cast}$ MR1$^{-/-}$) independent mice from three combined experiments (conventional T cells, CD4$^+$ T cells, CD8$^+$ T cells), two-tailed Mann–Whitney test. Numbers shown in MAIT cells panel indicate the proportion of MAIT cells (red gate) from TCRβ$^+$ gate (grey gate). **e** Absolute number and phenotype of NK cells (CD19$^-$TCRβ$^-$NK1.1$^+$) and CD11b/CD27 profile from lungs of B6-MAIT$_{cast}$ MR1 WT or B6-MAIT$_{cast}$ MR1$^{-/-}$ mice. Data is presented as mean ± SEM of $n = 19$ (B6-MAIT$_{cast}$ MR1 WT) and $n = 20$ (B6-MAIT$_{cast}$ MR1$^{-/-}$) independent mice from five combined experiments, two-tailed Mann–Whitney test (left) and $n = 6$ (B6-MAIT$_{cast}$ MR1 WT) or $n = 7$ (B6-MAIT$_{cast}$ MR1$^{-/-}$) independent mice from two combined experiments, Two-way ANOVA (right). **f** B16F10 tumor cells were inoculated into B6-MAIT$_{cast}$ MR1 WT or B6-MAIT$_{cast}$ MR1$^{-/-}$ mice as per **a** and NK cells depleted with anti-asialo GM-1 (αASGM1) on days -1 and 0. Data is presented as mean ± SEM of $n = 8$ (untreated B6-MAIT$_{cast}$ MR1 WT), $n = 9$ (untreated B6-MAIT$_{cast}$ MR1$^{-/-}$), $n = 6$ (anti-asialo GM-1 treated B6-MAIT$_{cast}$ MR1 WT) and $n = 5$ (anti-asialo GM-1 treated B6-MAIT$_{cast}$ MR1$^{-/-}$) independent mice and is representative of two independent experiments, One-way ANOVA. $*p < 0.05$, $**p < 0.01$, $***p < 0.001$, $****p < 0.0001$, ns = non-significant.

bearing mice at day 5 post-tumor challenge. This analysis revealed that while NK cell numbers were significantly increased in the lungs of B6-MAIT$_{cast}$ MR1$^{-/-}$ mice following tumor challenge, this effect was not observed in B6-MAIT$_{cast}$ MR1 WT mice (Supplementary Fig. 1h). Expression of the NK cell receptors DNAM-1, CD96 and NKG2D, the degranulation marker CD107a, and the cytokines IFN-γ and TNF, were similar in NK cells derived from tumor bearing B6-MAIT$_{cast}$ MR1 WT and B6-MAIT$_{cast}$ MR1$^{-/-}$ mice (Supplementary Fig. 1i). These results suggest that the enhanced anti-metastatic response observed in B6-MAIT$_{cast}$ MR1$^{-/-}$ mice is due to increased mature NK cell numbers.

**MAIT cell antigen pulsing of tumor cells reduces metastasis.** We next investigated the impact of MAIT cell activation on tumor metastasis by pulsing tumor cells with the MAIT cell antigen 5-OP-RU prior to intravenous inoculation (Fig. 2a). 5-OP-RU pulsing significantly upregulated MR1 on B16F10 melanoma cells (Fig. 2b), suggesting the tumor cells were able to present the 5-OP-RU antigen on their surface in an MR1-dependent manner. Furthermore, the number and fold-change of 5-OP-RU-pulsed tumor cell metastases were much lower than non-pulsed B16F10 tumors in B6-MAIT$_{cast}$ MR1 WT mice (Fig. 2c) and C57BL/6 WT mice (Fig. 2d). However, no reduction in tumor metastases following 5-OP-RU pulsing was observed in B6-MAIT$_{cast}$ MR1$^{-/-}$ mice (Fig. 2e), confirming the absolute requirement for MAIT cells in this protective effect. Similar results were obtained using a more stable but less potent 5-OP-RU derivative (JYM72, previously referred to as compound 11[38]) pulsed onto B16F10 tumor cells (Supplementary Fig. 2a). In these experiments, we also utilized a MSCV-cherry retroviral vector to overexpress MR1 on B16F10 cells (B16F10cherryMR1) and compared tumor growth to control B16F10 cells (B16F10cherry). MR1 overexpression resulted in an increase in cell surface MR1 expression, but did not significantly affect the extent of B16F10 metastases either in the presence or absence of antigen pulsing (Supplementary Fig. 2a–c). This suggests that endogenous MR1 expression on B16F10 tumor cells is sufficient for the anti-tumor response observed following pulsing with MAIT cell antigens. To confirm this, we evaluated the effect of 5-OP-RU pulsing on the metastasis of B16F10 tumor cells following CRISPR/Cas9-mediated deletion of MR1. Having verified successful knockout of

MR1 on B16F10 tumor cells (Supplementary Fig. 2d), we observed that 5-OP-RU no longer exhibited an anti-tumor effect (Fig. 2f), thus confirming the requirement for tumor-derived MR1 expression in this antigen-pulsing model. We also tested the effect of 5-OP-RU pulsing in the breast adenocarcinoma cell line E0771. In contrast to B16F10 cells, MR1 was significantly less upregulated in E0771 cells by 5-OP-RU (Fig. 2b), and accordingly the metastases of E0771 cells were unaffected by 5-OP-RU pulsing (Fig. 2g). These experiments demonstrate that two independent MAIT cell antigens, 5-OP-RU and JYM72, promote MAIT cell-dependent anti-tumor immunity when pulsed onto B16F10 tumor cells.

We next investigated whether the increased anti tumor immunity observed following 5-OP-RU pulsing was a direct mechanism, or an indirect mechanism involving downstream effector cells. Given our previous data indicating a dependence on NK cells in the protected phenotype of B6-MAIT$_{cast}$ MR1$^{-/-}$ mice, we performed additional experiments where NK cells were depleted using αASGM1. These experiments showed that there was no protection due to 5-OP-RU pulsing of B16F10 tumor cells following NK cell depletion (Fig. 2h), suggesting that MAIT cell-mediated activation of NK cells was required for the protective effect mediated by 5-OP-RU.

**In vivo expanded MAIT cells inhibit tumor metastasis.** Having established that activated MAIT cells enhance anti-tumor NK cell responses in the context of tumor cells presenting MAIT cell antigens, we next examined whether activation and expansion of MAIT cells could promote MAIT cell-mediated anti-tumor immunity when the antigen was not directly targeted to the tumor cells. To this end, 5-OP-RU was administered intranasally on days 0, 1, 2, and 4 prior to B16F10 tumor challenge on day 7 (Fig. 3a). 5-OP-RU treatment increased the frequency and number of MAIT cells within the lungs of B6-MAIT$_{cast}$ MR1 WT mice, with a sevenfold increase by day 6 compared to untreated mice (Fig. 3b and Supplementary Fig. 3a). 5-OP-RU treatment had little impact on the proportion of MAIT cells that were CD4$^+$, CD8$^+$, or double negative (DN), indicating that all subsets were expanded equivalently (Supplementary Fig. 3b). Activated MAIT cells also increased expression of the activation-associated protein, CD69, and NK activation receptor, DNAM-1, while their expression of the inhibitory ligand, CD96, was reduced (Fig. 3c). Having

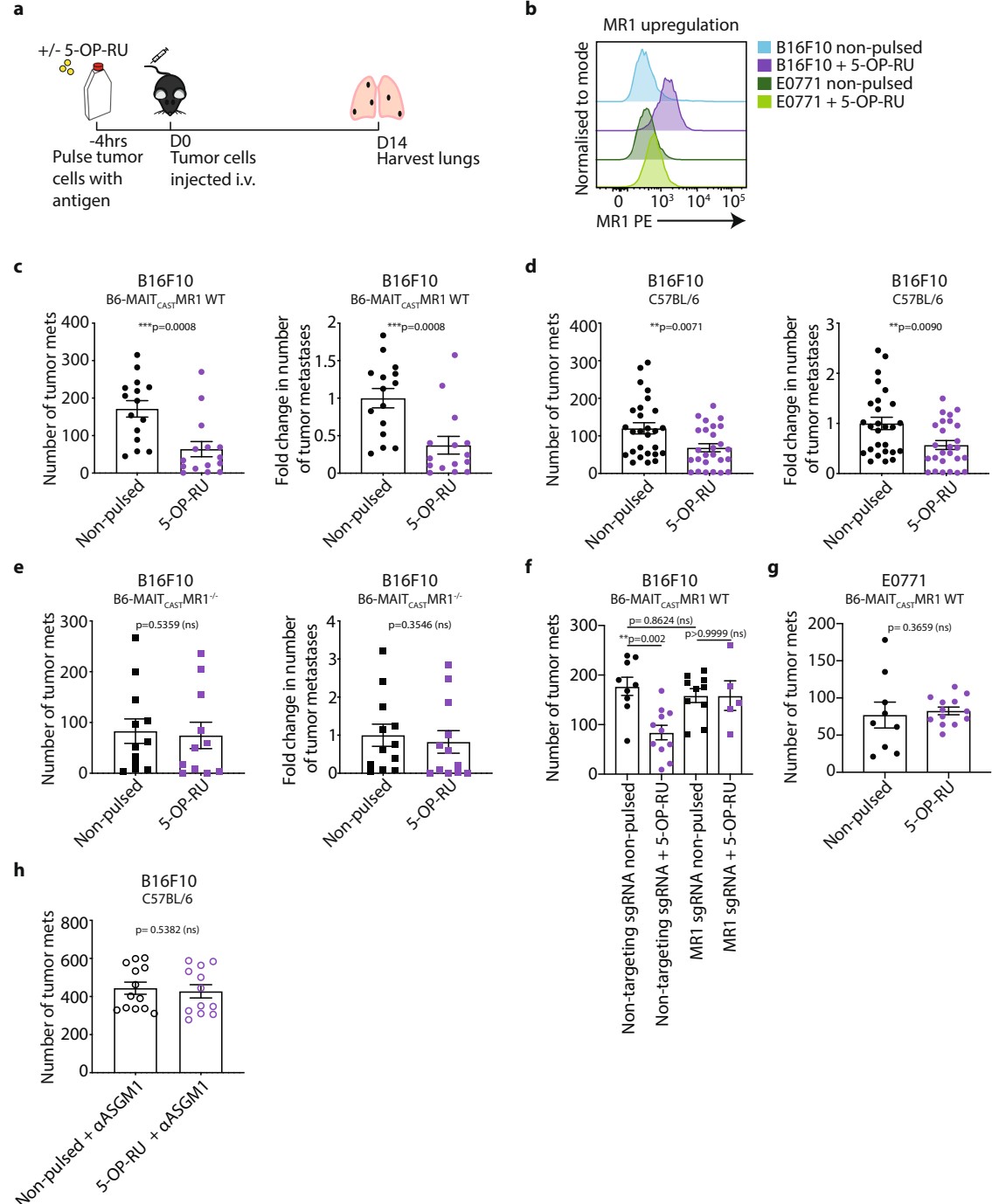

established that this treatment regimen successfully activated and expanded MAIT cells, we next challenged B6-MAIT_cast MR1 WT and B6-MAIT_cast MR1$^{-/-}$ mice with B16F10 tumors following MAIT cell expansion. Strikingly, MAIT cell expansion provided robust protection against B16F10 metastasis in B6-MAIT_cast MR1 WT mice (Fig. 3d), whereas no effect was observed in B6-MAIT_cast MR1$^{-/-}$ mice treated with the same 5-OP-RU-mediated expansion protocol (Fig. 3e), confirming the critical role of MAIT cells in this effect. Similar results were achieved when B6-MAIT_cast MR1 WT mice were challenged with E0771 tumors, confirming the effectiveness of MAIT cell expansion in multiple tumor models (Fig. 3f). Importantly, this suggests that it is not necessary for tumor cells to express MAIT cell antigens for MAIT cells to exert anti-tumor immune responses. Remarkably, in a therapeutic

model where E0771 tumors were administered prior to MAIT cell expansion in B6-MAIT_cast MR1 WT mice, significant protection was achieved (Fig. 3g, h). This therapeutic model highlights the potential of stimulating MAIT cells as a viable therapy for established cancers.

To investigate the mechanism of enhanced anti-tumor immunity following 5-OP-RU-mediated expansion, we performed gene expression analysis of lungs 5 days post 5-OP-RU treatment. This analysis revealed an increased expression of genes associated with MAIT cells, including *Rorc*, *Il17a* and *Cxcr6* (Supplementary Table 1 and Supplementary Fig. 4a), consistent with expansion of MAIT cells following 5-OP-RU treatment. Analysis of differentially expressed genes against the mouse gene atlas data repository revealed an enrichment in genes associated

**Fig. 2 Presentation of 5-OP-RU by tumor cells results in decreased metastasis in a MAIT and NK cell-dependent manner. a** Schematic of experimental setup. **b** Expression of MR1 in B16F10 and E0771 cells cultured with or without 10 μM of 5-OP-RU antigen for 4 h at 37 °C. Data is representative of three independent experiments. **c** B16F10 cells were cultured with or without 10 μM of 5-OP-RU antigen for 4 h at 37 °C and then $2 \times 10^5$ injected i.v. into B6-MAIT$_{cast}$ MR1 WT mice. Tumor metastases were enumerated 14 days after inoculation. Data is presented as mean ± SEM of $n = 15$ independent mice per group from two combined experiments, two-tailed Mann–Whitney test. **d** B16F10 cells were cultured with or without 10 μM of 5-OP-RU antigen for 4 h at 37 °C and $2 \times 10^5$ were injected i.v. into C57BL/6 mice. Tumor metastases were enumerated 14 days after inoculation. Data is presented as mean ± SEM of $n = 27$ (non-pulsed) and $n = 26$ (5-OP-RU-pulsed) independent mice from four combined experiments, two-tailed Mann–Whitney test. **e** B16F10 cells were cultured with or without 10 μM of 5-OP-RU antigen for 4 h at 37 °C and $2 \times 10^5$ were injected i.v. into B6-MAIT$_{cast}$ MR1$^{-/-}$ mice. Tumor metastases were enumerated 14 days after inoculation. Data is presented as mean ± SEM of $n = 12$ (non-pulsed) and $n = 11$ (5-OP-RU-pulsed tumor cells) independent mice from one independent experiment, two-tailed Mann–Whitney test. Data from Fig. 1a non-pulsed groups also present in **c** and **e**. **f** B16F10 non-targeting sgRNA and B16F10 MR1 sgRNA cells were cultured with or without 10 μM of 5-OP-RU antigen for 4 h at 37 °C and $2 \times 10^5$ were injected i.v. into B6-MAIT$_{cast}$ MR1 WT mice. Tumor metastases were enumerated 14 days after inoculation. Data is presented as mean ± SEM of $n = 9$ (non-targeting sgRNA non-pulsed), $n = 11$ (non-targeting sgRNA + 5-OP-RU), $n = 10$ (MR1 sgRNA non-pulsed) and $n = 5$ (MR1 sgRNA + 5-OP-RU) independent mice and is representative of two independent experiments, One-way ANOVA. **g** E0771 cells were cultured with or without 10 μM of 5-OP-RU antigen for 4 h at 37 °C and $5 \times 10^5$ were injected i.v. into B6-MAIT$_{cast}$ MR1 WT mice. Tumor metastases were enumerated 14 days after inoculation. Data is presented as mean ± SEM of $n = 9$ (non-pulsed) or $n = 13$ (5-OP-RU pulsed) independent mice representing two combined experiments, two-tailed Mann–Whitney test. **h** B16F10 cells were cultured with or without 10 μM of 5-OP-RU antigen for 4 h at 37 °C and $2 \times 10^5$ were injected i.v. into C57BL/6 mice. NK cells were depleted with anti-asialo GM-1 (αASGM1) on days –1 and 0. Tumor metastases were enumerated 14 days after inoculation. Data is presented as mean ± SEM of $n = 13$ (non-pulsed + αASGM1) or $n = 12$ (5-OP-RU-pulsed + αASGM1) independent mice representing two combined experiments, two-tailed Mann–Whitney test. **p < 0.01, ***p < 0.001, ns = non-significant.

with macrophages and NK cells, suggesting that MAIT cell expansion led to subsequent macrophage and NK cell recruitment, expansion and/or activation (Fig. 4a). To validate this, we investigated the immune infiltrate in the lungs of B6-MAIT$_{cast}$ MR1 WT mice by flow cytometry following intranasal expansion of MAIT cells with 5-OP-RU. This analysis revealed that 5-OP-RU mediated MAIT cell expansion resulted in increased numbers of macrophages (F4/80$^+$CD64$^+$ myeloid cells), which also exhibited an increased activation phenotype as shown by enhanced expression of MHCII (Fig. 4b). Moreover, we confirmed that 5-OP-RU-mediated MAIT cell expansion led to an increase in the absolute number of NK cells, particularly the mature CD11b$^+$CD27$^-$ subset in B6-MAIT$_{cast}$ MR1 WT mice (Fig. 4c). By contrast, the numbers of CD4$^+$ and CD8$^+$ T cells were unaffected by intranasal expansion of MAIT cells with 5-OP-RU (Fig. 4d). Notably, the increase in NK cells following 5-OP-RU intranasal administration was not observed in B6-MAIT$_{cast}$ MR1$^{-/-}$ mice, confirming the requirement of MAIT cells for this effect (Fig. 4c). A more comprehensive analysis of NK cell phenotype in B6-MAIT$_{cast}$ MR1 WT mice following 5-OP-RU administration revealed that NK cells expressed significantly increased levels of NKG2D, Ki-67 and KLRG1 (Fig. 4e), while other markers were unchanged. Given the known role of NKG2D in the elimination of tumor cells, including B16F10[39], the increased expression of NKG2D on NK cells following 5-OP-RU treatment may contribute to the enhanced NK cell-mediated protection of these mice.

As NK cells were previously shown to be critical for the protection against lung metastasis (Fig. 2h) and were expanded post 5-OP-RU administration in B6-MAIT$_{cast}$ MR1 WT mice (Fig. 4c), this led us to investigate the role of NK cell expansion in the protective effect of intranasal 5-OP-RU antigen stimulation. Similar to our results when tumor cells were pulsed with 5-OP-RU (Fig. 2h), NK cell depletion with αASGM1 ablated the protective effects of 5-OP-RU-mediated MAIT cell expansion (Fig. 4f). This was also observed following treatment with the NK1.1 targeting antibody (Supplementary Fig. 5a), demonstrating the important role for NK cells in mediating this effect. To further confirm this NK cell-mediated anti-tumor response, we also assessed the role of NK cells in MAIT cell-mediated tumor inhibition using NKp46$^{Cre}$-Mcl-1$^{loxp}$ mice, which lack functional NK cell development and reconstitution[40], and C57BL/6 WT control mice (Supplementary Fig. 5b). 5-OP-RU-mediated MAIT

cell expansion in these NKp46$^{Cre}$-Mcl-1$^{loxp}$ mice occurred to a similar extent as C57BL/6 WT mice controls (Supplementary Fig. 5c), but this had no effect on B16F10 metastasis in these mice that lack NK cells (Fig. 4g). Conversely, when conventional CD8$^+$ T cells were depleted, the protective effects of 5-OP-RU-mediated MAIT cell expansion were unaffected, suggesting CD8$^+$ T cells do not contribute to this anti-tumor response (Supplementary Fig. 5d). These results confirmed that MAIT cell-mediated activation of NK cells results in enhanced anti-tumor immunity, and that this does not require MAIT cell antigen presentation by tumor cells.

**MAIT cells enhance the anti-tumor function of NK cells in an IFN-γ-dependent manner.** To determine the molecular basis underlying the protective effects observed above, we isolated NK cells and MAIT cells from PBS- versus 5-OP-RU-treated B6-MAIT$_{cast}$ MR1 WT mice with and without B16F10 tumor challenge and performed RNA-seq. This analysis revealed that 5-OP-RU-mediated expansion of MAIT cells led to the upregulation of several genes associated with MAIT cell activation. Notably, we observed increased expression of *Ifng*, *Tnf* and *Gzmb* (granzyme B), but reduced expression of *Il17a*, by MAIT cells following 5-OP-RU-mediated expansion (Supplementary Fig. 6a). To assess the impact of MAIT cell activation on NK cells, we performed pathway analysis on differentially expressed genes in NK cells isolated from lungs of B6-MAIT$_{cast}$ MR1 WT mice treated with 5-OP-RU or PBS. Notably, IFN-γ response and inflammatory response pathways were prominent in this analysis, reflecting the activation of NK cells following 5-OP-RU treatment (Fig. 5a, b). These pathways were also enriched in NK cells isolated from tumor bearing B6-MAIT$_{cast}$ MR1 WT mice pre-treated with 5-OP-RU, highlighting that activation of these pathways is potentially associated with reduced metastatic burden. Additionally, genes associated with a type I IFN response and oxidative phosphorylation were also increased following 5-OP-RU activation (Fig. 5a and Supplementary Fig. 6b). Since type I IFNs[41] and oxidative phosphorylation[42] are associated with enhanced NK cell effector function, these data are also consistent with enhanced activation of NK cells following MAIT cell stimulation. Upregulation of genes associated with IFN-γ and IFN-α in NK cells post 5-OP-RU treatment was also consistent with enrichment of genes associated with the GO term "cytokine-mediated signaling pathway" following analysis of gene expression in whole lung tissue with the same expansion protocol

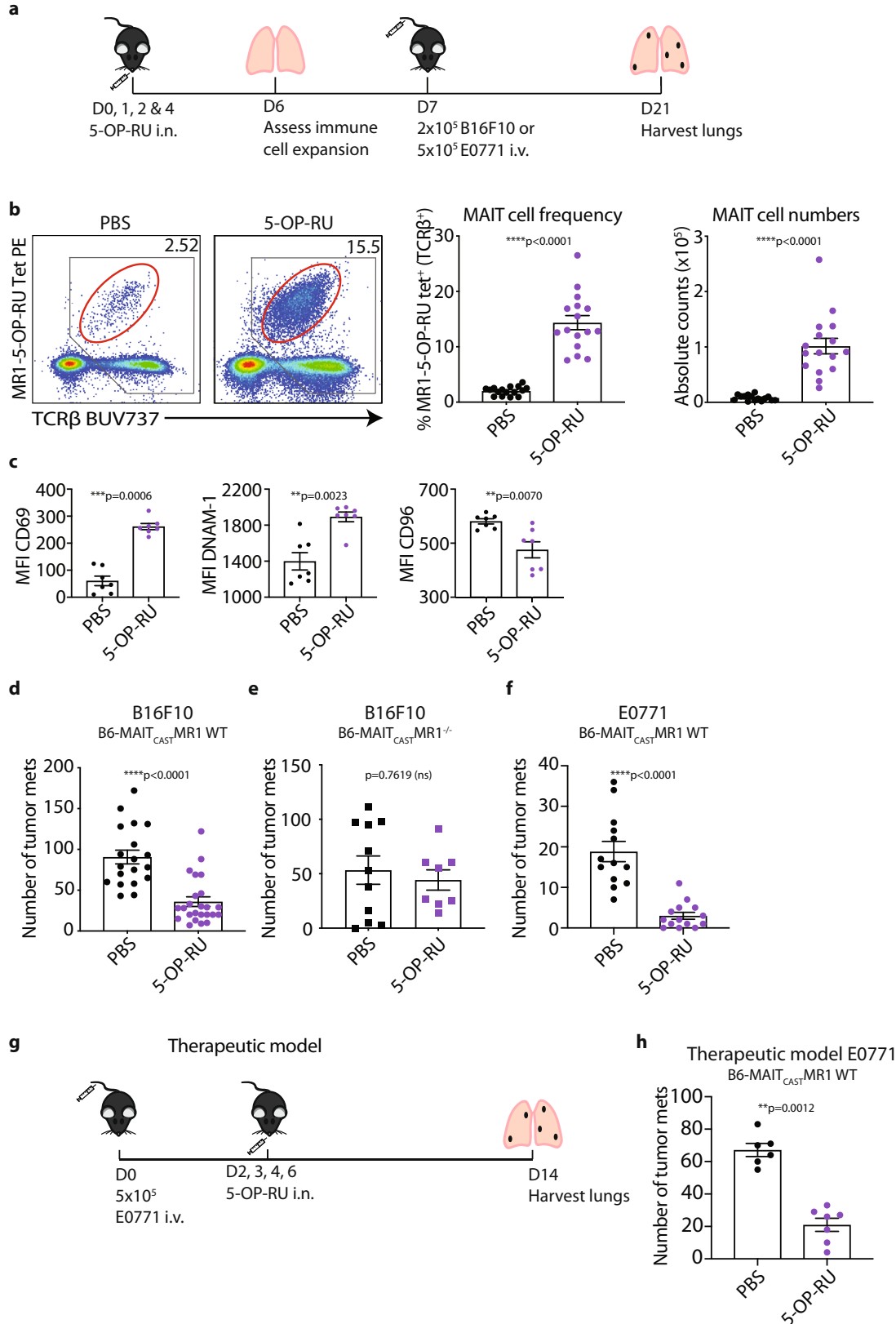

(Supplementary Fig. 4b, c). Given that this data suggested that IFN-γ-mediated activation of NK cells occurred following 5-OP-RU treatment, we next assessed the source of IFN-γ during this MAIT cell expansion protocol. Consistent with our RNA-seq analysis (Supplementary Fig. 6a), we observed increased IFN-γ and TNF production from MAIT cells (Fig. 5c), and this was observed at similar levels in all MAIT cell subsets (Supplementary Fig. 4d). By contrast, cytokine production from conventional CD4+ and CD8+ T cells was minimal after treatment with 5-OP-RU (Supplementary Fig. 4e). We next assessed the effect of IFN-γ blockade on 5-OP-

**Fig. 3 MAIT cell expansion via 5-OP-RU treatment results in enhanced anti-tumor immunity. a** Schematic of experimental setup. **b** Proportion and absolute number of MAIT cells at day 6 post treatment following intranasal administration of 50 μL of 233 μM 5-OP-RU or PBS at days 0, 1, 2, and 4 in B6-MAIT$_{cast}$ MR1 WT mice. Data is presented as representative flow cytometry plots and mean ± SEM of $n = 16$ independent mice per group from four combined experiments, two-tailed Mann–Whitney test. **c** Expression of indicated cell surface markers on MAIT cells following intranasal treatment of mice as per (**a**). Data is presented as mean ± SEM of $n = 7$ per group from two combined experiments, two-tailed Mann–Whitney test. **d** Following treatment of mice as per (**a**). $2 \times 10^5$ B16F10 tumor cells were injected i.v. into B6-MAIT$_{cast}$ MR1 WT mice. Tumor cell metastases were enumerated from lungs 14 days post tumor inoculation. Data is presented as mean ± SEM of $n = 19$ (non-treated) and $n = 24$ (5-OP-RU treated) independent mice from four combined experiments, two-tailed Mann–Whitney test. **e** Following treatment of mice as per (**a**). $2 \times 10^5$ B16F10 tumor cells were injected i.v. into B6-MAIT$_{cast}$ MR1$^{-/-}$ mice. Tumor cell metastases were enumerated from lungs 14 days post tumor inoculation. Data is presented as mean ± SEM of $n = 11$ (non-treated) and $n = 8$ (5-OP-RU treated) independent mice from two combined experiments, two-tailed Mann–Whitney test. **f** Following treatment of mice as per (**a**). $5 \times 10^5$ E0771 tumor cells were injected i.v. into B6-MAIT$_{cast}$ MR1 WT mice. Tumor cell metastases were enumerated from lungs 14 days post tumor inoculation. Data is presented as mean ± SEM of $n = 13$ (non-treated) and $n = 14$ (5-OP-RU treated) independent mice from two combined experiments, two-tailed Mann–Whitney test. **g** Schematic of experimental setup for therapeutic model. **h** $5 \times 10^5$ E0771 tumor cells were injected i.v. into B6-MAIT$_{cast}$ MR1 WT mice on day 0. Mice received 50 μL of 233 μM 5-OP-RU or PBS on days 2, 3, 4, and 6 and tumor cell metastases were enumerated from lungs 14 days post tumor inoculation. Data is presented as mean ± SEM of $n = 6$ (non-treated) and 7 (5-OP-RU treated) independent mice and is representative of two independent experiments, two-tailed Mann–Whitney test. $*p < 0.05$, $**p < 0.01$, $***p < 0.001$, $****p < 0.0001$, ns = non-significant.

RU-mediated protection. This revealed that IFN-γ was critical for the protective effect of 5-OP-RU treatment (Fig. 5d), which was concomitant with reduced 5-OP-RU-mediated expansion of NK cells following treatment with anti-IFN-γ (Supplementary Fig. 6c). Moreover, NK cells from 5-OP-RU pre-treated B6-MAIT$_{cast}$ MR1 WT mice exhibited an activated phenotype as shown by increased expression of genes associated with NK cell effector functions, such as *Ifng, Gzmb* and *Prf1* (perforin), as well as increased expression of the NK cell activation ligands, *Cd226* (DNAM-1) and *Ncr1* (NKp46) (Fig. 5e). These experiments highlight that activating MAIT cells can enhance NK cell effector functions and suggest that the combination of MAIT cell activation with therapies that promote NK cell activation may provide a novel approach to enhance NK cell-mediated anti-tumor immunity.

**Human MAIT-NK cell cross talk results in increased activation of NK cells.** To evaluate the effect of MAIT cells on NK cell-mediated anti-tumor immunity in human cancer, we developed a novel MAIT cell gene signature in order to investigate the prognostic significance of MAIT cells using publicly available TCGA datasets. To devise the signature, we performed differential gene signature analysis on MAIT cells versus 28 other immune cell types isolated from peripheral blood, based on a published dataset[43]. The genes identified from this analysis (Supplementary Table 2) were highly expressed in MAIT cells relative to other cell types including conventional T cells and NK cells (Supplementary Fig. 7a). Moreover, the ability of this gene signature to identify MAIT cells was comparable to the ability of a previously validated NK cell signature used to identify NK cells[44] (Supplementary Fig. 7b), thereby indicating that this gene signature can be used to evaluate the extent of MAIT cell infiltration in bulk tissues. We investigated the impact of the MAIT cell signature score on the progression-free survival (PFS) of patients with cancers where high levels of NK cells correlated with improved survival. The impact of a high NK score on prognosis was most marked in skin cutaneous melanoma (SKCM) and bladder cancer (BLCA) and in both these cancer types a high MAIT cell signature correlated with an increased NK cell signature and expression of CD3, CD8 and CD45 (Fig. 6a and Supplementary Fig. 8a) likely indicative of tumors with high levels of immune cell infiltrate. While the MAIT cell signature had no impact on PFS by itself (Supplementary Fig. 8b), a high MAIT score significantly reduced the impact of a high NK score on PFS (Fig. 6b). This suggests that, similar to the murine data presented above, MAIT cells can suppress NK cell-mediated anti-tumor immunity. Similar observations were found in triple-negative breast invasive carcinoma (BRCA TNBC),

cholangiocarcinoma (CHOL) and liver hepatocellular carcinoma (LIHC), where a high MAIT cell infiltration had a negative impact on the outcome of patients with a high NK cell infiltration (Supplementary Fig. 8c). These results imply that MAIT-cells may be detrimental to NK cell-mediated anti-tumor immunity in these patient cohorts.

To investigate whether human MAIT cell activation could activate human NK cells similar to our data in the murine system, we cultured peripheral blood mononuclear cells (PBMCs) from healthy donors in the presence or absence of 5-OP-RU. Stimulation of PBMCs with 5-OP-RU resulted in significantly increased production of IFN-γ and TNF by NK cells and to a much lesser extent conventional T cells (Fig. 6c and Supplementary Fig. 8d). To confirm that this was a MAIT cell-mediated effect, we isolated NK cells and Vα7.2$^+$ cells (enriched for MAIT cells) from healthy donors and cultured these cells alone or together in the presence or absence of 5-OP-RU. Stimulation of NK cells alone with 5-OP-RU did not result in increased production of IFN-γ. However, stimulation of NK cells cultured with MAIT cells and 5-OP-RU resulted in significantly increased production of IFN-γ by NK cells (Fig. 6d), confirming that 5-OP-RU-mediated stimulation of MAIT cells subsequently induces potent activation of NK cells. To assess whether 5-OP-RU-mediated activation of MAIT cells can prime NK cells for secondary stimulation, PBMCs were activated in the presence or absence of 5-OP-RU overnight and stimulated with either anti-CD16 or IL-12 and IL-18. Activation of NK cells with either anti-CD16 or IL-12 and IL-18 resulted in the production of IFN-γ and TNF. Importantly, this was significantly enhanced by prior stimulation of PBMCs with 5-OP-RU (Supplementary Fig. 8e). Moreover, 5-OP-RU activation of PBMCs led to enhanced cytotoxic function against K562 cells, a known NK cell target[45], inferring increased activation of NK cell cytotoxic function following 5-OP-RU treatment (Fig. 6e and Supplementary Fig. 8f). To investigate this in a patient context, we examined this effect using immune cells isolated from primary melanoma lesions. Owing to the low frequency of MAIT cells observed within these particular tumors, MAIT cells isolated from healthy donor PBMCs were added to melanoma samples with or without 5-OP-RU stimulation (Fig. 6f, g). Notably, the addition of MAIT cells and 5-OP-RU led to a significant increase in NK cell activation as measured by expression of CD69, IFN-γ and TNF (Fig. 6f, g). This suggests that activated tumor-infiltrating MAIT cells have the potential to enhance tumor-infiltrating NK cell function and thus promote an anti-tumor response. Notably, 5-OP-RU had no effect on NK cell function in the absence of exogenous MAIT cells, confirming the specificity of this cross-talk response (Fig. 6f, g). Since these effects were largely dependent on the addition of exogenous MAIT cells, we further explored the extent of endogenous MAIT-NK

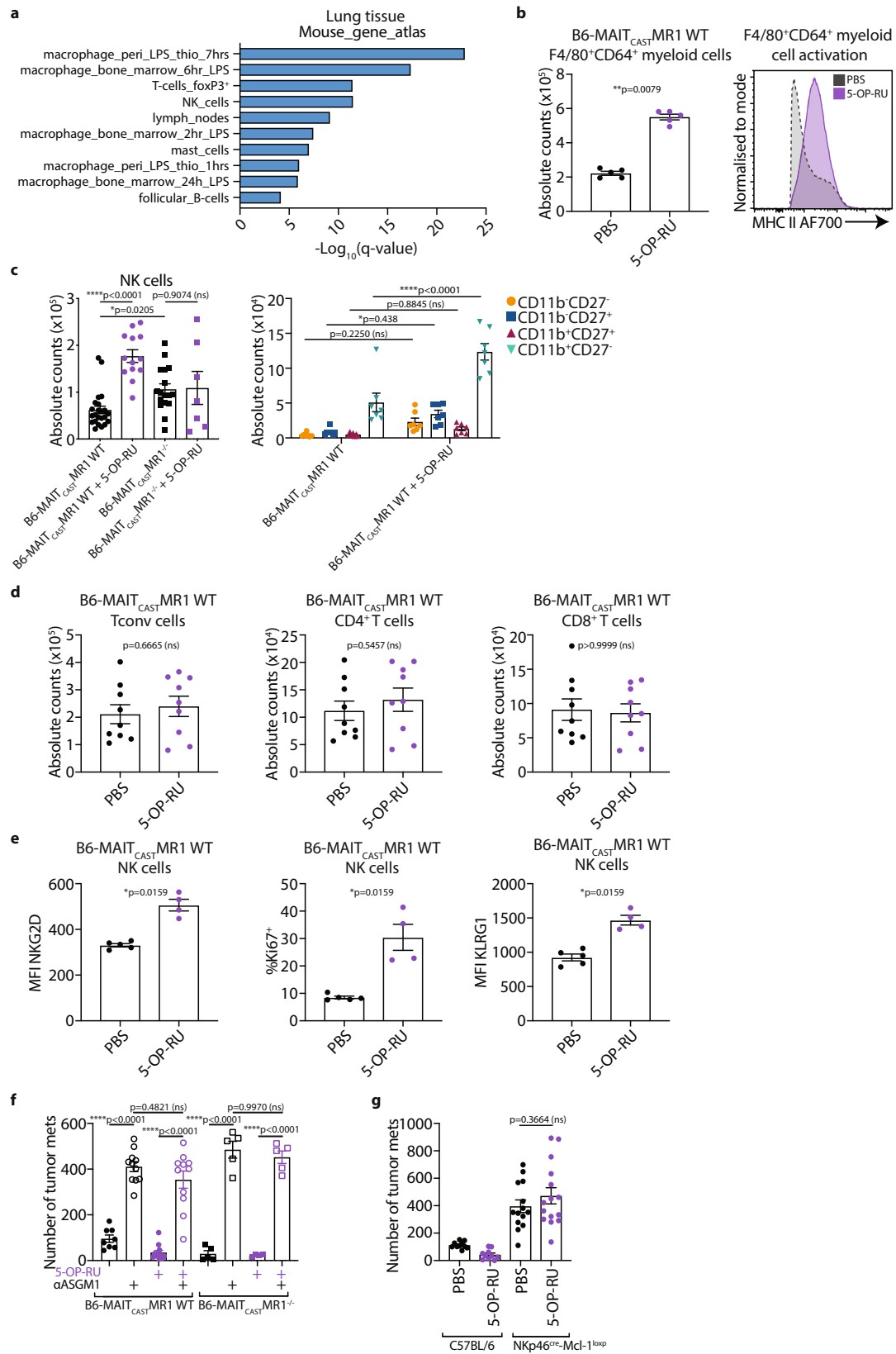

cell axis using a melanoma sample identified to have a higher MAIT cell frequency (0.78% of live cells, compared to <0.1% in Fig. 6f, g). 5-OP-RU treatment significantly increased the number of IFN-γ and TNF-positive NK cells, confirming that NK-MAIT cell cross talk can

occur within a tumor context (Fig. 6h). Taken together, this data suggests that in a patient setting MAIT cells can negatively impact NK cell anti-tumor functions. However, activation of MAIT cells can overcome this inhibition by enhancing NK cell function.

**Fig. 4 MAIT cell expansion results in enhanced NK cell activation and anti-tumor immunity. a** B6-MAIT$_{cast}$ MR1 WT mice were treated as per Fig. 3a. Lungs were harvested and RNA extracted from single-cell suspensions. Gene expression determined by Nanostring and differentially expressed genes with 5-OP-RU treatment compared to PBS controls were compared to the mouse gene atlas. **b** Absolute number and activation of F4/80$^+$CD64$^+$ myeloid cells from B6-MAIT$_{cast}$ MR1 WT mice treated as per Fig. 3a. Data is presented as mean ± SEM of $n = 5$ mice per group from one independent experiment, two-tailed Mann–Whitney test. **c** Absolute number of NK cells from B6-MAIT$_{cast}$ MR1 WT and B6-MAIT$_{cast}$ MR1$^{-/-}$ mice treated as per Fig. 3a (left) and CD11b and CD27 phenotype of NK cells in B6-MAIT$_{cast}$ MR1 WT treated with PBS or 5-OP-RU (right). Data is presented as mean ± SEM of $n = 23$ (non-treated B6-MAIT$_{cast}$ MR1 WT), $n = 13$ (5-OP-RU treated B6-MAIT$_{cast}$ MR1 WT), $n = 17$ (non-treated B6-MAIT$_{cast}$ MR1$^{-/-}$) and $n = 7$ (5-OP-RU-treated B6-MAIT$_{cast}$ MR1$^{-/-}$) independent mice from six combined experiments, One-way ANOVA (left). Data is presented as mean ± SEM of $n = 7$ independent mice per group from two combined experiments, representative of five independent experiments two-way ANOVA (right). Data from Fig. 1e. (left) also present in **c.** (left) present in **d** Absolute number of conventional T (Tconv), CD4$^+$ and CD8$^+$ T cells from B6-MAIT$_{cast}$ MR1 WT mice treated as per Fig. 3a. Data is presented as mean ± SEM of $n = 9$ independent mice per group from two combined experiments, two-tailed Mann–Whitney test. **e** Expression of NKG2D, Ki67 and KLRG1 on NK cells from B6-MAIT$_{cast}$ MR1 WT mice treated as per Fig. 3a. Data is presented as mean ± SEM of $n = 5$ (PBS treated) and $n = 4$ (5-OP-RU treated) independent mice from one independent experiment, two-tailed Mann–Whitney test. **f** B6-MAIT$_{cast}$ MR1 WT and B6-MAIT$_{cast}$ MR1$^{-/-}$ mice were treated as per Fig. 3a. NK cells were depleted with anti-asialo GM-1 (αASGM1) on days 6 and 7 and $2 \times 10^5$ B16F10 cells were injected i.v. on day 7. Tumor metastases were enumerated 14 days after inoculation. Data is presented as mean ± SEM of $n = 8$ (non-treated B6-MAIT$_{cast}$ MR1 WT), $n = 11$ (αASGM1 treated B6-MAIT$_{cast}$ MR1 WT), $n = 12$ (5-OP-RU treated B6-MAIT$_{cast}$ MR1 WT), $n = 11$ (αASGM1 and 5-OP-RU treated B6-MAIT$_{cast}$ MR1 WT), $n = 5$ (non-treated B6-MAIT$_{cast}$ MR1$^{-/-}$), $n = 5$ (αASGM1 treated B6-MAIT$_{cast}$ MR1$^{-/-}$), $n = 4$ (5-OP-RU treated B6-MAIT$_{cast}$ MR1$^{-/-}$) and $n = 5$ (αASGM1 and 5-OP-RU treated B6-MAIT$_{cast}$ MR1$^{-/-}$) independent mice from two combined experiments, one-way ANOVA. **g** C57BL/6 and NKp46$^{Cre}$-Mcl-1$^{loxp}$ mice were treated as per Fig. 3a and $2 \times 10^5$ B16F10 cells were injected i.v. on day 7. Tumor metastases were enumerated 14 days after inoculation. Data is presented as mean ± SEM of $n = 10$ (non-treated C57BL/6 and), $n = 10$ (5-OP-RU treated C57BL/6), $n = 14$ (non-treated NKp46$^{cre}$-Mcl-1$^{loxp}$) or $n = 15$ (5-OP-RU treated NKp46$^{cre}$-Mcl-1$^{loxp}$) independent mice from two combined experiments, one-way ANOVA. $^{**}p < 0.01$, $^{***}p < 0.001$, $^{****}p < 0.0001$, ns = non-significant.

## Discussion

While the role of MAIT cells in immunity to infection is well-established, their contribution to anti-tumor immune responses remains unclear. Some studies of MAIT cells in cancer patients have suggested that increased MAIT cell numbers within tumors is a negative prognostic factor, while others suggest that the presence of MAIT cells correlates with improved survival[9]. Our findings shed light on these paradoxical observations, revealing that while under steady state conditions MAIT cells negatively regulate NK cell maturation and NK cell-dependent anti-tumor immune responses. By contrast, activated and expanded MAIT cells promote anti-tumor immunity through IFN-γ-dependent activation of NK cells.

Our observations of MAIT cell-deficient mice being more resistant to B16F10 lung metastasis and subcutaneous tumor growth suggested that MAIT cells interfere with effective anti-tumor immunity. As MAIT cell-deficient mice showed increased NK cell number and enhanced maturation status, and the protective effect in MAIT cell-deficient mice was lost when NK cells were depleted, we propose the existence of a MAIT-NK cell axis that controls NK cell-mediated tumor immunity. Importantly, we have also provided evidence that a similar MAIT-NK cell axis may exist in humans given that a strong MAIT cell signature abolished the otherwise improved prognosis associated with a strong NK cell signature. The observation that MAIT cell-deficient mice have enhanced NK cells is reminiscent of a previous report showing that NKT cell-deficient mice have increased levels of MAIT cells[46]. It is possible that MAIT cells actively control NK cell function via the production of inhibitory cytokines or other factors, such as IL-13, which was recently shown to be abundantly produced by MAIT cells following chronic stimulation[24].

Importantly, despite the apparent suppressive role MAIT cells naturally play in cancer, we have also shown that, paradoxically, activation and in vivo expansion of MAIT cells can be harnessed to promote NK cell-mediated anti-tumor immunity. NK cells were critical effectors of this 5-OP-RU-mediated therapeutic effect, but interestingly 5-OP-RU administration also led to an increase in number and activation of macrophages. Although our study does not exclude a direct role for macrophages in this anti-tumor effect, our data more strongly point to a role for NK cells given that effects were totally abrogated following NK cell depletion. We therefore hypothesize that macrophages may contribute to the therapeutic anti-tumor effect through chemokine-mediated recruitment of NK cells as has been reported previously[47].

Mechanistically, RNA-seq-based experiments showed that MAIT cell activation led to an increase in NK cell genes associated with an IFN-γ response. The importance of IFN-γ in these MAIT- NK cell-mediated anti-tumor effects was supported by MAIT cell production of IFN-γ following in vivo activation, and confirmed by the complete abrogation of protection following IFN-γ neutralization. This is reminiscent of previous studies showing that activated CD1d-restricted NKT cells also promote anti-tumor immunity via the activation of NK cells in an IFN-γ-dependent manner[48, 49]. In these studies IFN-γ production by NKT cells led to a subsequent activation of NK cells and in turn increased secretion of IFN-γ by NK cells themselves. It is possible that a similar mechanism occurs following MAIT cell activation whereby the effects of MAIT cell-derived IFN-γ are subsequently amplified by the production of IFN-γ by NK cells or other immune cells. Our observation that activated MAIT cells enhance NK cell anti-tumor responses contrasts with a recent study, which indicated that MAIT cell activation through 5-OP-RU pulsing of tumor cells led to suppression of NK cell anti-tumor immunity[50]. Reasons for these differences are not clear, however, we achieved similar MAIT cell dependent activation of NK cells when 5-OP-RU was presented by either tumor cell pulsing or via intranasal delivery. Moreover, our observations were confirmed in B6-MAIT$_{cast}$ MR1 WT mice that have MAIT cell numbers more equivalent to humans, and were not observed in B6-MAIT$_{cast}$ MR1$^{-/-}$ mice, which lack MAIT cells. In addition to the mouse-based studies, we also showed that MAIT cell activation could enhance NK cells derived from human tumors. Interestingly, another recent study found that tumor-derived antigen presentation by MR1 to conventional T cells resulted in significant anti-tumor activity[51]. Taken together, this suggests that activating MAIT cells may be a potential therapeutic strategy for the treatment of cancer.

It is well accepted that effector functions of tumor-infiltrating conventional T cells can be suppressed by the TME. Likewise, human studies have shown reduced cytotoxic functions of MAIT cells in mucosal-associated cancers[29–31]. Strategies to overcome this decreased effector function are critical for effective

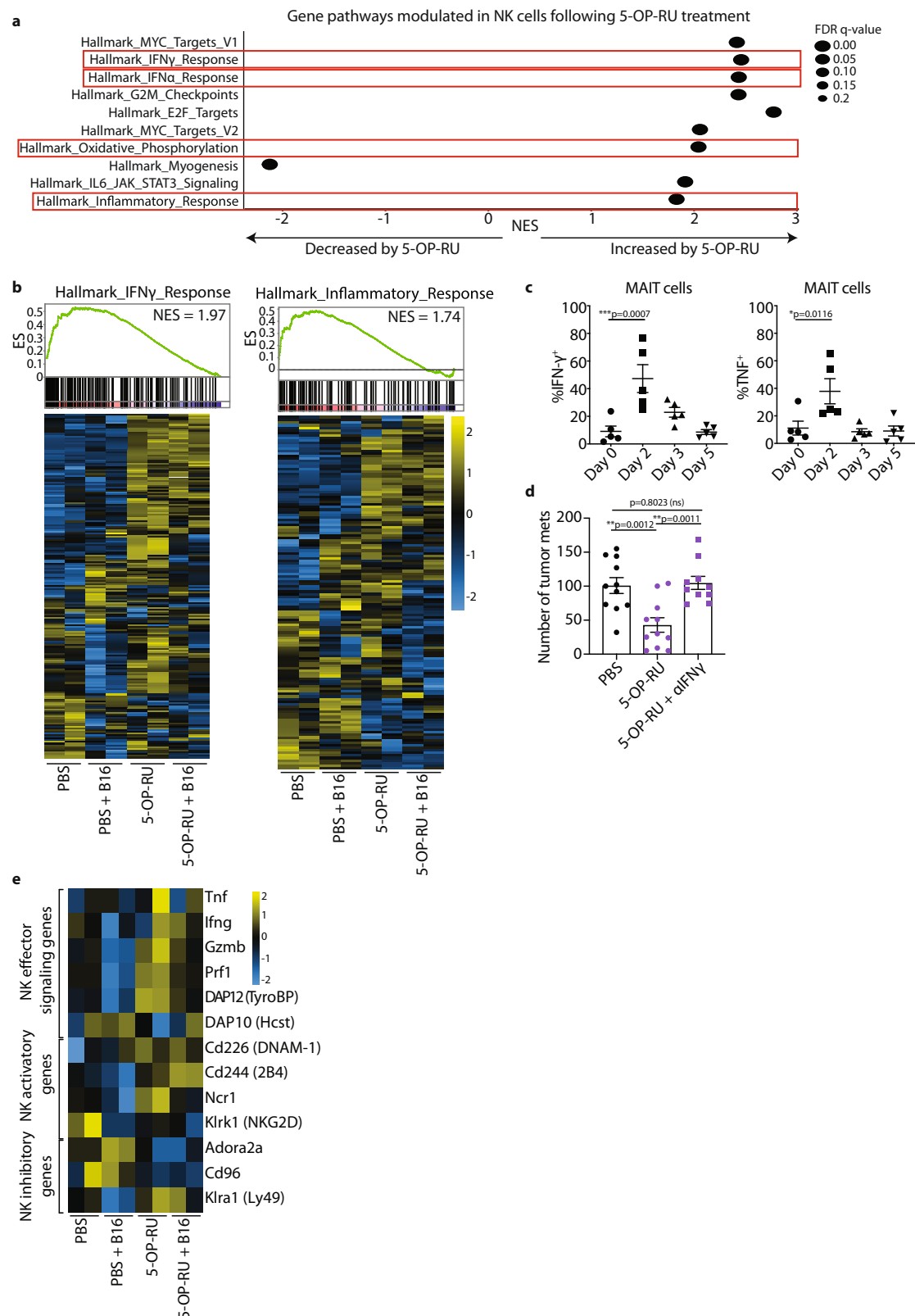

immigrotherapies. Excitingly, our data has revealed two distinct avenues by which MAIT cells may be targeted for tumor immunotherapy. If we can effectively deplete or block MAIT cells in steady state, this may release MAIT cell inhibitory effects over NK cells. Conversely, antigen-specific activation of MAIT cells in

cancer patients has the potential to reverse this diminished effector response, and ultimately lead to improved NK cell-mediated tumor rejection. Given that MAIT cells all exhibit the same specificity in humans, and MR1 is a non-polymorphic antigen presenting molecule that presents the same antigens in all

**Fig. 5 MAIT cell-mediated enhancement of NK cell anti-tumor function is IFN-γ dependent. a, b** B6-MAIT$_{cast}$ MR1 WT mice were treated as per Fig. 3a. $5 \times 10^5$ B16F10 cells or PBS control were injected i.v. at day 6 and 12 h later lungs were processed. NK1.1$^+$ NK cells were FACS sorted and RNA extracted for subsequent 3′ RNA-sequencing analysis. Cells were harvested from three mice per group and data represented as technical duplicates. Heatmaps show relative expression of genes in indicated pathways. **a** Differentially expressed genes were subject to hallmark pathway analysis. Indicated signatures were significantly upregulated post intranasal 5-OP-RU treatment. **b** Heatmaps showing expression of indicated genes in pathways identified. **c** B6-MAIT$_{cast}$ MR1 WT mice were treated as per Fig. 3a. Expression of IFN-γ and TNF from MAIT cells at indicated days post 5-OP-RU expansion. Data is presented as mean ± SEM of $n = 5$ independent mice per group from one independent experiment, one-way ANOVA. **d** C57BL/6 mice were treated as per Fig. 3a, inoculated with $2 \times 10^5$ B16F10 cells and received 250 μg of anti-IFN-γ i.p. on day 0 and 1. Data is presented as mean ± SEM of $n = 11$ (non-treated and 5-OP-RU treated) and $n = 10$ (5-OP-RU + αIFN-γ treated) independent mice, from two combined experiments, one-way ANOVA. **e** Heatmaps showing expression of indicated genes in pathways identified from Fig. 5b. $*p < 0.05$, $**p < 0.01$, $****p < 0.0001$, ns = non-significant.

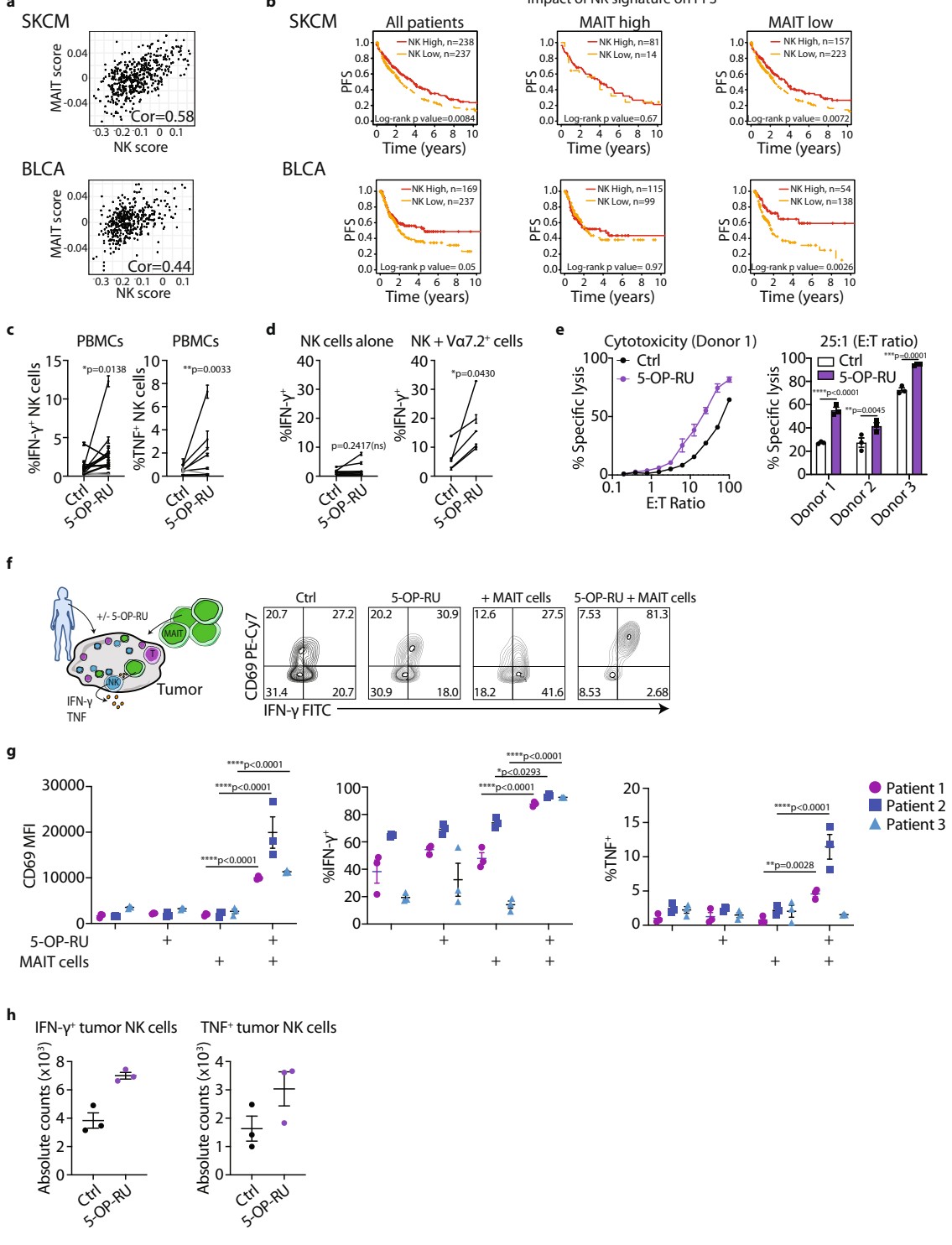

**Fig. 6 Activated Human MAIT cells cross talk with NK cells leading to increased activation, cytokine production and cytotoxicity. a** Correlation (cor) between MAIT score and NK score as indicated for skin cutaneous melanoma (SKCM) and bladder carcinoma (BLCA). **b** Analysis of progression-free survival (PFS) of SKCM and BLCA patients stratified into indicated cohorts based upon high or low expression of NK and MAIT cell related genes. **c** PBMCs were cultured for 16 h with IL-2 (20 IU/mL) in the presence or absence of 100 nM 5-OP-RU. Production of cytokines by $CD3^-CD56^+$ NK cells determined by flow cytometry. One donor (gray) was identified to have a low frequency of MAIT cells ($0.80\%$ $CD19^-CD3^+CD161^+V\alpha7.2^+$ of live cells). Data is presented as mean ± SEM of triplicate cultures from 17 independent donors ($IFN-\gamma^+$) or triplicate cultures from 8 independent donors ($TNF^+$), two-tailed Paired-t test. **d** Isolated NK cells and MAIT ($V\alpha7.2^+$) cells were cultured for 16 h with IL-2 (20 IU/mL) in the presence or absence of 100 nM 5-OP-RU. Production of cytokines by $CD3^-CD56^+$ NK cells determined by flow cytometry. Each data point represents the mean ± SEM of triplicate cultures from nine independent donors (NK cells only) or triplicate culture from five independent donors (NK cells + $V\alpha7.2^+$ cells), two-tailed Paired-$t$-test. **e** PBMCs were stimulated as in (**c**) and co-cultured with $Cr^{51}$-labeled K562 target cells. PBMC: tumor ratios (E:T ratio) are as indicated. Data is represented by mean ± SEM of triplicate cultures from a representative experiment of $n = 3$. Percentage-specific lysis at 1:25 PBMCs: tumor targets. Data is represented by mean ± SEM of triplicate cultures from three separate donors, two-way ANOVA. **f, g** Tumor infiltrating lymphocytes from melanoma patients were cultured with IL-2 (20 IU/mL) in the presence or absence of 100 nM 5-OP-RU with or without additional MAIT cells isolated from healthy donor PBMCs. Expression of CD69 and percentage of IFN-γ or TNF-producing NK cells ($CD3^-CD56^+$) assessed by flow cytometry. Data is presented as mean ± SEM of triplicate cultures from three independent tumor samples, two-way ANOVA. **h** Tumor infiltrating lymphocytes from a melanoma tumor sample were cultured as in (**c**) and expression of IFN-γ and TNF on NK cells were assessed. Data is presented as mean ± SEM of triplicate cultures. $*p < 0.05$, $**p < 0.01$, $***p < 0.001$, $****p < 0.0001$, ns = non-significant.

individuals, these approaches present fewer obstacles than targeting conventional MHC-restricted T cells that express highly diverse TCRs and interact with highly polymorphic antigen presenting molecules. In summary, our data reveals a critical MAIT cell-NK cell axis, that can both inhibit and promote NK cell-mediated immunity in different settings. The therapeutic potential of these findings needs to be explored especially in mucosal-associated cancers, which have high levels of NK and MAIT cells[9].

## Methods
**Cell lines and mice**. The C57BL/6 mouse melanoma cell line B16F10 and the human chronic myelogenous leukemia cell line K562 were obtained from ATCC. The breast adenocarcinoma cell line E0771 was obtained from Prof. Robin Anderson (Olivia-Newton John Cancer Centre Heidelberg, Victoria, Australia). Tumor lines were verified to be mycoplasma negative by PCR analysis. MR1 overexpressing B16F10 cells were generated using the MSCV–cherry retroviral vector[37]. Tumor cells were grown in DMEM supplemented with 10% FCS, glutamax and penicillin/streptomycin. For in vivo experiments, the indicated number of cells were resuspended in PBS and injected subcutaneously (100 μL) or intravenously (200 μL) into C57BL/6 WT, B6-MAIT$_{cast}$ MR1 WT[33], B6-MAIT$_{cast}$ MR1$^{-/-}$[33] or NKp46$^{Cre}$-Mcl-1$^{loxp}$ mice[52], which were all bred separately. All experiments used sex-matched mice aged 6–12 weeks that were housed in PC2-specific pathogen-free conditions at the Peter MacCallum Cancer Centre (Melbourne). Mice were assessed for rapid-breathing or other signs of illness every 1–2 days. Subcutaneous tumors were measured with callipers and survival was defined when tumor size exceeded 100 mm$^2$. Lung metastases were individually counted with a dissection microscope. Mice were euthanized via cervical dislocation or $CO_2$ asphyxiation at the end of the experiment. All animal experiments were approved by the Animal Experimental Ethics Committee (Peter MacCallum Cancer Centre; Protocol E582).

**Antigens, antibodies, and recombinant cytokines**. The MR1 ligands 5-(2-oxoepropylideneamino)-6-D-ribitylaminouracil (5-OP-RU) and JYM72 were synthesized following published procedures[38]. The compounds were diluted in PBS immediately prior to in vivo administration. Neutralizing antibodies for IFN-γ (H22), CD8 (YTS 169.4) and NK1.1 (PK136) were purchased from BioXcell (West Lebanon, NH, USA). Anti-asialo GM-1 (αASGM1) was purchased from Wako Chemicals (Richmond, VA, USA). Recombinant IL-2, IL-12, and IL-18 were purchased from PeproTech (Rehovot, Israel). Additional information for antibodies and reagents can be found in Supplementary Data 1.

**Pulsing of tumor cells**. Tumor cells were pulsed with 10 μM of 5-OP-RU or JYM72 for 3 h in DMEM supplemented with 10% FCS, glutamax and penicillin/streptomycin at 37 °C before i.v. injection into mice. Expression of MR1 was assessed via flow cytometry to ensure presentation of antigen by tumor cells.

**Intranasal expansion of MAIT cells**. Mice were anaesthetized with xylazine and ketamine prior to receiving 50 μL of 233 μM 5-OP-RU diluted in PBS into the nasal cavity on day 0, 1, 2, and 4. Expansion of MAIT cells in the lung was subsequently determined via flow cytometry analysis.

**Analysis of lung-infiltrating immune subsets**. Lungs were excised and digested post-mortem with 1 mg/mL collagenase type IV (Worthington; Lakewood, NJ, USA) and 0.01 mg/mL DNase (Sigma-Aldrich Pty Ltd, Sydney, NSW, Australia). After digestion at 37 °C for 30 min at 120 rpm, red blood cells (RBCs) were lysed using ACK lysis buffer and cells were passed through a 70 μM filter twice. Anti-CD16/32 mAb (clone 2.4G2) was used to block Fc receptors prior to flow cytometry staining.

**Flow cytometry**. For immunofluorescence staining, morphology of cells was determined via FSC-A and SSC-A, doublets excluded via FSC-A and FSC-H and dead cells excluded with Fixable-Yellow live/dead stain (Invitrogen, Carlsbad, CA, USA). Human and mouse MR1 tetramers were produced[13, 53]. Biotinylated MR1-5-OP-RU and control MR1-Ac-6FP monomers were tetramerized with streptavidin conjugated to PE (BD Pharmigen; San Jose, CA, USA). For intracellular detection of IFN-γ and TNF, cells were cultured for 4 h with 5 ng/mL PMA and 1 μg/mL Ionomycin in the presence of Golgi Plug and Golgi Stop (BD Pharmingen; San Jose, CA, USA). Data were collected using a BD FACSymphony and analyzed using FacsDiva v8 and FlowJo v10.2 (Treestar, Ashland, OR). A list of antibodies used for flow cytometry is provided in Supplementary Table 3and gating strategies are shown in Supplementary Fig. 9.

**CRISPR/Cas9-mediated knockout of MR1**. To generate the MR1 knockout B16F10 cell line (B16F10 MR1 sgRNA), 37 pmoles sgRNA (Synthego) and 270 pmoles recombinant Cas9 (IDT) were incubated together for 10 min at room temperature. sgRNA/Cas9 RNPs were electroporated into $3 \times 10^5$ B16F10 cells using a Lonza SF Cell line kit and 4D-Nucleofector. MR1 knockout was confirmed via flow cytometric analysis. Two sgRNAs were used for MR1 knockout (G1: CGACAGUGUCACUCGACAGA, G2: AGGUACACUCAGCUGCUAAG). B16F10 non-targeting sgRNA control cell lines were also generated (G1: GCACUACCAGAGCUAACUC).

**Nanostring gene expression analysis**. RNA prepared from whole lung tissue was quantified and quality assessed using Tapestation analysis. Gene counts were determined by Peter MacCallum Cancer Centre Advanced Genomic Core by nCounter XT assay (Mouse Pan Cancer Immune Profiling; Nanostring technologies 115000142) and analyzed by nSolver 4.0 software as per manufacturer's instructions. All positively upregulated genes were tested in ENRICHR tool[54, 55]. Top enriched genesets within the Mouse gene atlas[56] and Gene Ontology (GO) Biological process 2018 database[57] were identified, including macrophage and NK cell genesets[56] and Cytokine mediated signaling pathway (GO:0019221). Heatmaps for the top 100 most differentially expressed genes and genes from the cytokine mediated signaling pathway were generated using the pheatmap R package, using row-mean centered and scaled-gene expression of log2(normalized counts + 0.5).

**RNA-sequencing analysis**. MAIT cells and NK cells were sorted with a BD FACSAria, and RNA was isolated using a RNeasy mini kit (Qiagen) as per the manufacturer's instructions. RNA-seq libraries were prepared using the QuantSeq 3′ mRNA-seq Library Prep Kit for Illumina (Lexogen) as per manufacturer's instructions. Single-end, 75 bp RNA-seq short reads were generated using NextSeq (Illumina, Inc., San Diego, CA). CASAVA 1.8.2 was used for base calling. Quality of the data was assessed using RNA-SeQC v1.1.7[58]. To analyze differential gene expression, the data was quality trimmed using Cutadapt v1.6 to remove random primer bias and 3′ end trimming was performed to remove poly-A-tail derived reads and alignment performed using HISAT2 against the mouse reference genome

mm10. The subread software package 1.6.4 was used to count the number of reads per gene using gene definitions from Ensembl Release 96[59]. Gene counts were normalized using the Voom method in the Limma R package and converted into log2 counts per million. Linear modeling using Limma was applied for differential expression analysis between defined groups[60]. Adjusted p-values were computed using the Benjamini–Hochberg method. All differentially expressed genes were filtered for false-discovery rate (FDR) cutoff of 5% and fold-change cutoff >1. Volcano plots were used to represent differential gene expression between groups. Heatmaps for selected gene sets were generated using the pheatmap R package, using row-mean centered and scaled-gene expression levels of log2(CPM + 0.5). Rows were grouped by hierarchical clustering using Euclidean distance and average-linkage. For gene set enrichment analysis (GSEA), all differentially expressed mouse genes between NK cells in 5-OP-RU-treated mice against PBS controls were ranked according to fold-change and the GSEA v3.0 software (http://www.broad.mit.edu/gsea/) was used to test for enrichment to hallmark genesets in the molecular signatures database (MSigDB)[61].

**Isolation of peripheral blood mononuclear cells (PBMCs), NK cells, MAIT cells, and cytotoxicity assays.** Human buffy coats from healthy blood donors were provided by the Australian Red Cross Blood Service with written informed consent and approval from Melbourne Human Ethics Committee (01/14). PBMCs were isolated via standard density gradient using Ficoll-paque Plus (GE Healthcare; Chicago, IL, USA) and RBCs were lysed with ACK lysis buffer. PBMCs were cultured in complete RPMI media with 20 IU/mL recombinant human IL-2 with or without 100 nM 5-OP-RU. NK cells were isolated using the human NK cell isolation kit (Miltenyi Biotec). MAIT cells were isolated via PE conjugated anti-Vα7.2 and anti-PE Microbeads (Miltenyi Biotec). PBMC killing function was assessed in a 4 h $^{51}$Cr-release assay against K562 targets at E:T ratios (PBMCs: tumor targets) of 100:1, 50:1, 25:1, 12.5:1, 6.25:1, 3.13:1, 1.56:1, 0.78:1, and 0.39:1. Cytotoxicity was normalized to maximum killing for the experiment.

**MAIT and NK signature generation.** To calculate a gene signature associated with MAIT cells, RNA-Seq profiling of 29 different PBMCs were downloaded from the Gene Expression Omnibus (Accession ID GSE107011). Differential expression was performed using limma-voom between the MAIT cell group and each other cell type. A MAIT cell gene set was calculated by selecting the genes that had nominal p-value < 0.05 and log-fold-change > 0 against all other cell types. For NK cells, an established gene set was used[44]. KLRB1 was excluded from the signatures due to being in both original gene sets. Signature scores were calculated using the singscore R package[62]. Heatmaps for GSE107011 were generated from gene-wise Z-scores of log-counts-per-million using the Complex-Heatmap R package[63].

**Survival analysis.** Samples were classified into "High" and "Low" categories for MAIT and NK cell groups based on whether a sample had a signature score above or below (respectively) the mean value of that cell type's signature scores within that TCGA cohort. Progression-free survival (PFS) endpoints were matched to samples from an integrated TCGA pan-cancer clinical data resource[64]. Kaplan–Meier and Cox regression models on different subsets of the TCGA data using the "survival" R package (version 3.1-8).

**Preparation of melanoma biopsy cell suspension.** Patients undergoing surgical resection of melanoma metastases were enrolled in a prospective protocol after approval from the Melanoma Research Victoria Human Research Ethics Committee (07/38). Melanoma Research Victoria performed the informed consent for collection of samples for research purposes and all participants signed and agreed. Melanoma TILs were prepared as previously described[65].

**Statistical analysis.** Figures were generated using GraphPad Prism v7 and 8 (GraphPad Software). Data was first tested for normal distribution. Statistical differences were analyzed as stated in text. p < 0.05 was considered significant.

**Reporting summary.** Further information on research design is available in the Nature Research Reporting Summary linked to this article.

## Data availability

The nanostring data, RNA-sequencing data and MAIT and NK cell signatures that support the findings of this study have been deposited in GEO NCBI under the accession codes GSE179140, GSE179139, and GSE107011, respectively. Hallmark datasets utilized can be accessed via https://www.gsea-msigdb.org/gsea/msigdb/index.jsp. Source data are provided with this paper.

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

## Acknowledgements

The authors would like to acknowledge the assistance of the Animal Facility technicians at the Peter MacCallum Cancer Centre and Prof. Olivier Lantz for the provision of B6-MAITcast mice. This work was funded by Program Grants from the National Health and Medical Research Council (NHMRC; 1132373, 1113293, and 1140406) and a Cancer Council Victoria grant (APP1143517). F.S.F.G received a grant #1158085 awarded through the Priority-driven Collaborative Cancer Research Scheme and funded by Cure Cancer Australia with the assistance of Cancer Australia. P.A.B. was supported by National Breast Cancer Foundation Fellowship (ID# ECF-17-005, 2016-2020) and a Victorian Cancer Agency Mid-Career Fellowship (MCRF20011, 2021–current). I.G. House is supported by a Victorian Cancer Agency Early Career Fellowship (ECRF20017). P.K.D. is supported by an NHMRC Senior Research Fellowship (APP1136680). D.I.G. and F.P.F. are supported by NHMRC Senior Principal Research Fellowships (1117766, 1117017). H.-F.K. is supported by an NHMRC ECF Fellowship (1160333); D.I.G., H-F.K., J.Y.W.M., and D.P.F. are also supported by the Australian Research Council CE140100011.

## Author contributions

P.A.B, P.K.D., and D.I.G. conceived the idea. P.A.B., P.K.D., D.I.G., and H.-F.K. supervised the study. E.V.P., M.A.H., K.S., K.L.T., J.Lai, I.G.H., J.Li, A.X.Y.C., A.J.O., J.M., A.J. F., L.G., J.D.C., T.R.M., F.S.F.G. performed the experiments. E.V.P., S.P.K., M.Z. analyzed the data. A.P., J.Y.W.M., T.R.M., F.S.F.G., C.J.K., R.M., R.G.R., N.D.H., J.M., J.O., D.P.F., P.J.N. contributed with key reagents. E.V.P., P.A.B., P.K.D., D.I.G., and H.-F.K. wrote the manuscript. All authors discussed the results and reviewed the manuscript.

## Competing interests

J.M. and D.P.F. are named inventors on a patent application (PCT/AU2013/000742, WO2014005194) and J.Y.W.M., J.M. and D.P.F. are named inventors on another patent application (PCT/AU2015/050148, WO2015149130) involving MR1 ligands for MR1-restricted MAIT cells owned by University of Queensland, Monash University and University of Melbourne. A.P. current position is supported by Roche-Genentech. N.D. H. is co-founder, shareholder, and advisor of oNKo-Innate Pty Ltd. P.J.N. has research funding from BMS, Roche Genentech, Compugen, and Allergan. F.S.F.G. is a consultant and has funded research agreements with Biotheus Inc. P.K.D. declares the following conflicts: research funding from Myeloid Therapeutics, Prescient Therapeutics and Juno Therapeutics. P.A.B. declares the following conflicts: research funding from AstraZeneca, Bristol-Myers Squibb, and Gilead Sciences. The remaining authors declare no competing interests.
