## [Peer Review File · Nature Communications]

Reviewers' comments:

Reviewer #1 (Cancer immunity, NK biology) (Remarks to the Author):

MAIT cells regulate NK cell mediated tumor immunity

The present work of Petley et al. hypothesize that MAIT cells regulate NK cell activity in responses against tumors. More specifically they claim that naïve MAIT cell negatively regulate NK cells, while activated MAIT cells positive regulate NK cells.

The work can be subdivided in three parts:

Naïve MAIT negative regulate NK (part 1), while activated MAIT positively regulate NK via IFN γ (part2), and that this is similar in humans (part 3).

Comments

The authors claim that MAIT cells from the congenic mice (naïve or activated) can regulate NK cell activity.

While the hypothesis of the authors is interesting, their data do not support it. Large part of this is due to the use of B6-MAITCAST mice in most experiments. B6-MAITCAST mice harbor unusual numbers of MAIT cells, meaning unusual for this genetic background.

As pointed out by the authors, B6-MAITCAST mice harbor up to 20-fold more MAIT cells than wtB6 mice.

This might amplify any functional impact of that cell population by the same factor. And this might give a function to that cell type it would normally not have.

a) MAIT cells in the B6-MAITCAST mice might only acquire the ability to regulate NK cells, because the increased number results in an increase in total amount of factors produced by these cells that then suffice to achieve NK cell regulation while the "normal" level of those factors in B6 would not achieve that.

b) It is equally possible that MAIT cells in congenic B6 mice are found in anatomic localizations where they are normally not present, because their physiologic niche is inaccessible. Possibly, in such a case they could acquire the ability to regulate NK cells which they would not have in a wtB6 mouse.

Therefore, it is not clear if the observations made represent physiologic activities of MAIT cells.

- How compare B6-MAITCAST MAIT cells to wtB6 MAIT cells functionally?
- The authors need to provide data for wtB6 controls as well as wtB6 MR1^{-/-} controls that support their findings (wtB6 seem to have a sizable number of lung MAIT cells according to the authors data).

To the potential impact on NK cells:

! Strategy used by the authors to identify NK cells is insufficient: CD19-CD3-NK1.1+. Lung also contain ILC1 that share this phenotype. Since NK/ILC1 have not been analyzed so far in the context of these B6-MAITCAST congenic mice, a clear identification of NK vs ILC1 is necessary: CD49a and CD49b is not enough, as these markers distinguish NK from ILC1 only in the liver. The authors need to stain for the transcription factor Eomes to unambiguously identify NK cells.

As B6-MAITCAST mice have more MAIT cells how does that affect the NK cell compartment compared to wtB6 in the lung?

- Numbers of lung NK cells in B6-MAITCAST mice vs wtB6 and the ratio NK cells vs ILC1 in those mice

What is already very clear is, that the CD27 vs CD11b profile of NK cells from B6-MAITCAST mice is not equivalent to those from wtB6 mice. In the latter, almost all ($\geq 90\%$) have the CD11b+CD27- phenotype (which is different from the spleen). In the B6-MAITCAST mice only 40-50% (depending on the presence of MR1) are of that subset.

- Are there other differences – receptor expression, effector molecules? RNAseq ?
- Are there also functional differences between NK cells from B6-MAITCAST mice compared to wtB6?

Furthermore, the direct link between MAIT cell activation and NK cell activation via IFN γ from the former cells is not clearly shown neither the essential role of MAIT cell-derived IFN γ for the subsequent anti-B16F10 response of NK cells.

IFN γ is produced by many cell types and is has pleiotropic activities on many cell types. Two of those have been shown to be activated upon 5-OP-RU treatment in this work: macrophages and NK cells. Macrophages have not been further studied, yet, are known to regulate NK cell activity. It is conceivable that MAIT cell-derived IFN γ activates macrophages that in turn activate NK cells.

Several reports have demonstrated the essential role of NK cells in the protection against B16F10 lung metastases. Depleting NK cells in the current set of experiments (B6-MAITCAST and mice B6-MAITCAST MR1 $^{-/-}$ mice treated or not with 5-OP-RU) does only show their essential role, but not the involvement of MAIT cells. The authors need to show that specifically MAIT cell-derived IFN γ is important for the activity of NK cells against B16F10 metastases.

Also, the cytotoxic properties of activated MAIT cells or MAIT cells against MAIT-antigen pulsed tumor cells needs to be shown. It is announced in the text (Line 191/192), but no data shown. Please, show the data.

Importantly, no data are provided and no hypothesis is put forward to explain the negative regulation of NK cells by naïve MAIT cells (mouse or human).

General remark:

The authors state that B6-MAITCAST Mr1 $^{-/-}$ mice lack MAIT cells, but only provide 1 plot in Supplemental Figure 1A. Published data suggest that MAIT cells of the lung might only be mildly affected by the MR1-deficiency (Cui, JCI)

The authors need to provide solid data to prove that in their hands B6-MAITCAST Mr1^{-/-} mice lack MAIT cells in the lung. This is an essential point for most experiments.

This MS contradicts an earlier report that activated MAIT cells would inhibit NK cells in a similar model of lung metastases. While the authors cite that work in the discussion and mention that the reason for the discrepancy is not clear, they fail to recognize that the published report is based on work in B6, while the current MS is based on work with congenic B6 that has increased numbers of MAIT.

Also, the designation of B6-MAITCAST mice as "MR1" and B6-MAITCAST MR1^{-/-} mice as "MR1^{-/-}" in the figures is misleading. Especially, as the authors often refer to the congenic B6-MAITCAST mice simply as "mice" in the text. As such it is not immediately clear if the data are derived from B6 or B6-MAITCAST mice. The authors need to clearly identify B6 and mice throughout the text and figures (not only figure legends)!

Specific comments

Figure 1A, B, E, F: comparison to wtB6 necessary

- How compare NK cells from B6-MAITCAST and mice B6-MAITCAST MR1^{-/-} mice functionally? Are the former less active: Killing of RMA-S, YAC-1, cytokine production?
- What is the mechanism of negative regulation?

Figure 3

In vivo expanded MAIT cells inhibit tumor metastasis

- have "normal" wtB6 in vivo expanded MAIT cells the same effect?
- Too few info on E0771 tumors: growth in B6-MAITCAST Mr1^{-/-} not shown.

Figure 4

Is 5-OP-RU specifically activating only MAIT cells?

- Test of the same Lot of 5-OP-RU used in the experiments in the paper on purified NK cells in vitro

Figure 4E: B6-MAITCAST + 5-OP-RU vs untreated B6-MAITCAST Mr1^{-/-} = same outcome on number of metastases : how to explain? Also seen for wtB6 and wtB6 MR1^{-/-}?

The observed increase of MAIT cell genes revealed mainly MAIT17 (IL-17 producers) genes Rorc, Il17a, Cxcr6: not MAIT1 (IFN γ) specific genes though. Comments?

Macrophages are found to be expanded and activated just like NK cells, but not followed up by the authors:

- what cytokines do the secreted macrophages produce? IFN γ ? IL-12, IL-15, IL-18?

The increase of inflammatory response genes by NK cells from 5-OP-RU treated B6-MAITCAST mice supports the idea that inflammatory macrophages might activate NK cells.

Figure 5

Kinetic of IFN γ production by MAIT cells after treatment with 5-OP-RU: Day 2 is rather late effect:

- if gating on IFN γ + cells at this point, which cell types come up?
- Are NK cells already activated at day1 ?

Figure 5D: flaw in setup: IFN γ from NK cells is essential for response to B16; blocking IFN γ in this setting shows probably only that. Any impact of MAIT cell-derived IFN γ cannot be interpreted from this setup. Source of IFN γ not clear.

Related: Supplemental Figure 6C: source of IFN γ not clear

5F: A2AR $^{-/-}$ adding this mutant is only distracting. Role of that molecule for NK cell anti-tumor response appear unrelated. Should be removed.

Figure 6

Human MAIT-NK cell cross talk results in increased activation of NK cells

Line: 308-310: statement is based on what? And why would that be good or important?

Figure 6B: SKCM, BRCA TNBC: high MAIT score do not seem to reduce impact of a high NK cell score, but reduced the difference between impact of NK hi and NK lo scores by improving the impact of NK low scores!

BLCA, CHOL: hi MAIT improves impact of both NK hi and lo scores

Figure 6C: purified NK cells +/- MAIT cells should be used

Figure 6E-G: legend in plots and diagrams misleading: according to the figure legend IL-2 is present in all settings

Figure 6G actually indicates that 5-OP-RU has an effect on NK cells as it alone (on IL-2 preactivated cells) can induce IFN γ production from around 60% of NK cells – just like IL-2. But what is missing here is the control, i.e., PBMC prepared as before but cultured without IL-2 for the last 4hrs.

Along the same line as before, a MAIT-NK cross talk cannot be deduced from the experimental data as PBMCs and not purified populations were used.

Minor:

178: onto not into

Line 206: 333WT mice? Should be B6-MAITCAST mice

Reviewer #2 (Innate immunity, innate lymphoid cells, MAIT cell) (Remarks to the Author):

The authors describe the interplay between MAIT and NK cells in tumors. They show that in the absence of MAIT cells mice are less susceptible to tumor growth and metastases in the B16 model and the phenotype is dependent on NK cells, suggesting that MAIT cells negatively regulate NK cell numbers, maturation and anti-tumor activity. In line with these findings, a MAIT gene signature in patients negatively impacted the prognostic significance of NK cells. In contrast, the authors show that antigen-activated MAIT cells provide protection in murine models and promote NK cell activation in both murine models and in vitro experiments using tumor samples from patients. The authors propose that MAIT cells may be either beneficial or immunosuppressive in tumors depending on the functional state and antigen-specific activation.

The paper is potentially interesting but, at this stage, some of the data is confusing. While the different MAIT cell-dependent regulation of NK cells in the two settings is novel and interesting, the mechanism for this should be clarified. Thus, the main concept of this paper should be better demonstrated. Two opposite effects of MAIT cells in tumors are described. In both cases MAIT cell impact is dependent on NK cell regulation but drives contrasting outcomes in the modulation of tumor immunosurveillance. The divergent role of MAIT cells in tumors (in particular mucosal tumors) has already been described, showing that MAIT cells can be pathogenic producing IL-17 and IL-22 or protective having a type 1 signature. How are MAIT cells differently affecting NK cell effector functions in the two experimental settings?

Specific comments are listed below.

- MR1-deficient mice are protected from tumor growth in the B16 model and this is dependent on NK cell activation (Figure 1). A more careful analysis of NK cell activation (cytokine production, cytotoxicity markers) would be helpful. How are MAIT cells negatively impacting on NK cell activation in this context?
- Antigen-activated MAIT cells are beneficial in tumor models and this is still dependent on NK cells but having the opposite effect (Figure 2). How do the authors explain that no difference was observed in the presence or absence of MAIT cells using tumors pulsed with a MAIT specific antigen (Figure 2C)? It looks like the effect is not dependent on MAIT cells.
- The authors show that MR1 upregulation per se does not impact the phenotype observed (Supplementary Figure 2). They show no effect using the E0771 cell line where they say that MR1 is not upregulated. MR1 is actually absent in this case (Figure 2B). Maybe the antigen pulsing did not work or MR1 is not expressed at all in this cell line. I think this experiment should be revised and more clearly explained.
- In the Supplementary figure 2, the authors also show that tumor growth is not impacted in the absence of antigen pulsing comparing B16cherry cells and B16MR1-overexpressing cells (last panels). Actually, with the antigen there is no difference between the two cell lines either (first panels). Why are the authors now using B16 cherry cells?
- Intranasal exposure to a specific MAIT antigen is sufficient to promote tumor control (Figure 3). How do the authors explain that the E0771 model works in this case? If the tumors still do not express MR1 it means that MAIT cell activation fully occurred in the lung and it is sufficient to provide protection in the tumor.
- NK cell activation is the proposed mechanism underlying the reduced tumor growth (metastases) upon antigen-driven activation of MAIT cells in both the mouse model (Figures 4-5) and in vitro

experiments using human cells (Figure 6). How is this occurring? In the in vitro experiments (Figure 6), is the effect of 5-OP-RU dependent on MAIT cells?

- The analysis of the MAIT cell gene signature and the correlation with clinical data in TCGA cohorts is interesting and clear but the methods should be clarified.
- The analysis between multiple groups should be performed and a proper statistical test (Anova) should be used throughout the paper (Figure 1F, Figure 2C, Supplementary Figure 2...).

Minor points:

- Figures layout and panel numbering should be improved. Figures letters are missing and it was very complicated to follow the text.
- Is the quantification of MAIT and NK cell activation done in tumor bearing mice? A proper quantification of the markers and statistical analysis should be performed. (Supplementary Figure 1).
- B16 s.c. experiment (Supplementary Figure 1) in MR1-deficient mice with or without NK cells depletion would be informative.

Reviewer #1 (Cancer immunity, NK biology) (Remarks to the Author):

Comments

1) The authors claim that MAIT cells from the congenic mice (naïve or activated) can regulate NK cell activity. While the hypothesis of the authors is interesting, their data do not support it. Large part of this is due to the use of B6-MAITCAST mice in most experiments. B6-MAITCAST mice harbor unusual numbers of MAIT cells, meaning unusual for this genetic background.

As pointed out by the authors, B6-MAITCAST mice harbor up to 20-fold more MAIT cells than wtB6 mice. This might amplify any functional impact of that cell population by the same factor. And this might give a function to that cell type it would normally not have.

Response: We are pleased that the reviewer agrees that our hypothesis is interesting. We believe that our data fully supports this hypothesis and unfortunately the reviewer seems to have overlooked some of the key data in our paper where our observations of MAIT cell regulation of NK cell activity in B6-MAIT_{cast} mice were indeed confirmed using C57BL/6 WT mice (**Figure 2D, Figure 2H, Figure 4G, Supplementary Figure 5A-B**).

Furthermore, with regard to the reviewer's negative comments on the use of the B6-MAIT_{cast} mice model, we believe that this criticism is not justified given we confirmed all observations in both B6-MAIT_{cast} and C57BL/6 WT mice. Moreover, the frequency of MAIT cells in the lungs of B6-MAIT_{cast} mice is more similar to the frequencies observed in humans, and indeed, this was one of the motivations for their original generation. The frequency of MAIT cells in the lungs of B6-MAIT_{cast} mice (approximately 2-3%) is only 2-3 times higher than observed in C57BL/6 WT mice. The figure of 20-fold difference reported in the Cui *et al.* study (PMID 26524590) is based upon V α 19-J α 33 mRNA expression in these mice strains. In our study, we used 5-OP-RU loaded tetramers to identify MAIT cells, which is now widely regarded as the most definitive way to identify MAIT cells. Therefore, our method of MAIT cell detection was much more accurate.

As mentioned above, the frequency of MAIT cells in B6-MAIT_{cast} mice more closely resembles that observed in humans. For example, the frequency of MAIT cells in human lungs is 2-4% (PMID 7588203) and so is more in line with what is observed in B6-MAIT_{cast} mice as opposed to C57BL/6 WT mice. We therefore believe that B6-MAIT_{cast} mice represent a superior model to C57BL/6 WT mice for the purpose of evaluating the therapeutic activity of MAIT cells but again we reiterate that findings were recapitulated in C57BL/6 WT mice in any case.

2) MAIT cells in the B6-MAITCAST mice might only acquire the ability to regulate NK cells, because the increased number results in an increase in total amount of factors produced by these cells that then suffice to achieve NK cell regulation while the "normal" level of those factors in B6 would not achieve that.

Response: Please refer to our response to point 1 above. Given that the numbers of MAIT cells more closely recapitulate the human setting we believe the use of this model is of high relevance and complements our observations in the C57BL/6 WT mice

3) It is equally possible that MAIT cells in congenic B6 mice are found in anatomic localizations where they are normally not present, because their physiologic niche is inaccessible. Possibly, in such a case they could acquire the ability to regulate NK cells which they would not have in a wtB6 mouse.

Response: Please refer to our response to point 1 above. Whilst there is no reason to believe that MAIT cells are located in distinct anatomical locations, indeed this was investigated by the Cui *et al.* study, our studies in the C57BL/6 WT mice discount this as a possible confounding factor in our observations.

4) How compare B6-MAITCAST MAIT cells to wtB6 MAIT cells functionally?

Response: Please refer to our response to point 1 above. Our observations that MAIT cells can regulate NK cell activity in B6-MAIT_{cast} mice were confirmed using C57BL/6 WT mice (**Figure 2D, Figure 2H, Figure 4G, Supplementary Figure 5A-B**).

5) The authors need to provide data for wtB6 controls as well as wtB6 MR1^{-/-} controls that support their findings (wtB6 seem to have a sizable number of lung MAIT cells according to the authors data).

Response: Please refer to our response to point 1 above.

To the potential impact on NK cells:

6) Strategy used by the authors to identify NK cells is insufficient: CD19-CD3-NK1.1⁺. Lung also contain ILC1 that share this phenotype. Since NK/ILC1 have not been analysed so far in the context of these B6-MAITCAST congenic mice, a clear identification of NK vs ILC1 is necessary: CD49a and CD49b is not enough, as these markers distinguish NK from ILC1 only in the liver. The authors need to stain for the transcription factor Eomes to unambiguously identify NK cells.

Response: To address this concern we have performed additional experiments with the inclusion of the Eomes marker. Using the gating strategy of CD45.2⁺CD19⁻TCRβ⁻NK1.1⁺Eomes⁺ to define NK cells, we observed that NK1.1⁺ cells expressed Eomes, confirming that NK1.1⁺ cells were *bona fide* NK cells. This data supports our original conclusions and is shown in **Supplementary Figure 1C** and referred to in the following text on line 152:

“Using a more comprehensive gating strategy to discriminate NK1.1⁺ conventional NK cells from ILC1s, we confirmed that these NK1.1⁺ cells expressed Eomes, confirming their identity as NK cells (**Supplementary Figure 1C**).”

7) As B6-MAITCAST mice have more MAIT cells how does that affect the NK cell compartment compared to wtB6 in the lung? Numbers of lung NK cells in B6-MAITCAST mice vs wtB6 and the ratio NK cells vs ILC1 in those mice.

Response: We have performed additional experiments comparing the number of NK cells in the lung of B6-MAIT_{cast} and wtB6 (B6-MR1 WT) mice and show no significant difference in the number of NK cells. This data is shown below as a **Reviewer Only Figure**. In addition, we would like to reiterate that our observations were consistent between B6-MAIT_{cast} and wtB6 mice in any case. With that

point clarified, we hope that the reviewer will agree that there is no need to speculate that NK cell numbers are altered in the B6-MAIT_{cast} mice.

Reviewer Only Figure 1: NK cell numbers are comparable in B6-MR1 WT and B6-MAIT_{cast} MR1 WT mice. Lungs from B6-MR1 WT, B6-MAIT_{cast} MR1 WT and B6-MAIT_{cast} MR1^{-/-} mice were harvested and NK cells were analysed via flow cytometry. Data is presented as mean ± SEM of n = 9 - 11, One-way ANOVA, *p < 0.05, ***p < 0.001, ns = non-significant.

8) What is already very clear is, that the CD27 vs CD11b profile of NK cells from B6-MAITCAST mice is not equivalent to those from wtB6 mice. In the latter, almost all (≥ 90%) have the CD11b⁺CD27⁻ phenotype (which is different from the spleen). In the B6-MAITCAST mice only 40-50% (depending on the presence of MR1) are of that subset.

Response: We are unsure of the basis for this comment since we did not compare the expression of CD11b and CD27 on NK cells derived from B6-MAIT_{cast} and wtB6 mice in our original manuscript. However, to address this concern, we have performed new experiments that show the proportion of NK cells that are CD11b⁺CD27⁻ is mildly increased in wtB6 mice relative to B6-MAIT_{cast} mice (75% vs. 60%). This is consistent with the hypothesis that MAIT cells restrict the more mature NK subset in the steady state and so is consistent with our original conclusions. This data is shown below as **Reviewer Only Figure 2.**

Reviewer Only Figure 2: Proportion of mature NK cells in B6-MR1 WT mice, B6-MAIT_{cast} MR1 WT and B6-MAIT_{cast} MR1^{-/-} mice. Lungs from naïve B6-MR1 WT, B6-MAIT_{cast} MR1 WT and B6-MAIT_{cast} MR1^{-/-} mice were harvested and NK cell expression of CD11b and CD27 was analysed via flow cytometry. Data is presented as mean ± SEM of n = 4 - 5, Two-way ANOVA, **p<0.01, ***p<0.001.

9) Are there other differences – receptor expression, effector molecules? RNAseq ?

Response: We do not believe that a major undertaking of RNA-seq analysis between B6-MAIT_{cast} mice and wtB6 mice is warranted as the relevant control for the B6-MAIT_{cast} MR1 WT mice is the B6-MAIT_{cast} MR1^{-/-} mice. Again we reiterate that our findings were consistent between the B6-MAIT_{cast} and wtB6 strain (Please refer to our response to point 1 above). With that point clarified, we trust that the reviewer will agree that any potential differences between these strains do not affect our conclusions.

10) Are there also functional differences between NK cells from B6-MAITCAST mice compared to wtB6?

Response: To address this question we have performed additional experiments comparing the cytotoxic activity of B6-MAIT_{cast} and wtB6 NK cells. These experiments reveal that there is no major difference in the cytotoxic activity of NK cells from these strains of mice. Considering that this is not a study to compare B6-MAIT_{cast} and wtB6 mice, we feel that this comparison will be a distraction from the key messages of the paper. As mentioned in previous comments, with the clarification that the observed effects apply to both B6-MAIT_{cast} and wtB6 mice, we hope that the reviewer will agree that it isn't necessary to focus on how different NK cells might be between these two strains.

This new data is shown below as a **Reviewer Only Figure 3** but can be included in the manuscript if requested.

Reviewer Only Figure 3: Cytotoxicity of NK cells from B6-MAIT_{cast} MR1 WT and B6-MR1 WT mice is comparable. Splenocytes from naïve B6-MAIT_{cast} MR1 WT and B6-MR1 WT mice were harvested and cultured in 1000 IU/mL IL-2 for 5 days. Cells were then cocultured with Cr⁵¹-labelled B16F10 target cells. NK cells (Effector): tumor (Target) ratios (E:T ratio) are as indicated. Data represented as the mean ± SEM of triplicate cultures from 5 pooled mice.

11) Furthermore, the direct link between MAIT cell activation and NK cell activation via IFN γ from the former cells is not clearly shown neither the essential role of MAIT cell-derived IFN γ for the subsequent anti-B16F10 response of NK cells. IFN γ is produced by many cell types and is has pleiotropic activities on many cell types. Two of those have been shown the be activated upon 5-OP-RU treatment in this work: macrophages and NK cells. Macrophages have not been further studied, yet, are known to regulate NK cell activity. It is conceivable that MAIT cell-derived IFN γ activates macrophages that in turn activate NK cells..... The authors need to show that specifically MAIT cell-derived IFN γ is important for the activity of NK cells against B16F10 metastases.

Response: Whilst, the reviewer suggests an interesting experiment, it would require the use of a transgenic mouse with specific knockout of IFN γ within the MAIT cell compartment and, to our knowledge, such a mouse model unfortunately does not exist. However, to show the direct modulation of NK cells by MAIT cells is direct we have performed additional experiments where MAIT cells were cocultured with NK cells in the absence of other cell types. This data indicates that MAIT cell stimulation leads to NK cell activation in this context, confirming a direct link between these two cell types. Please refer to point 7 in response to Reviewer 2 above. It is entirely possible that MAIT cell activation leads to the production of IFN γ that subsequently leads to the activation of NK cells and other cell types to secrete their own IFN γ , thus creating an amplification effect. However the key first step is MAIT cell activation since no NK cell expansion is seen in MAIT cell deficient mice (**Figure 4C**). We have added additional discussion on this topic on line 423:

“In these studies IFN γ production by NKT cells led to a subsequent activation of NK cells and in turn increased secretion of IFN γ by NK cells themselves. It is possible that a similar mechanism

occurs following MAIT cell activation whereby the effects of MAIT cell-derived IFN γ are subsequently amplified by the production of IFN γ by NK cells or other immune cells.”

12) The cytotoxic properties of activated MAIT cells or MAIT cells against MAIT-antigen pulsed tumor cells needs to be shown. It is announced in the text (Line 191/192), but no data shown. Please, show the data.

Response: We believe the reviewer is referring to the following statement: “We next investigated whether the increased anti tumor immunity observed following 5-OP-RU pulsing was due to increased MAIT cell mediated cytotoxicity, or an indirect mechanism involving downstream effector cells.” This statement refers to the potential *in vivo* cytotoxicity of MAIT cells as a hypothetical mechanism and does not infer direct cytotoxic activity as this was not shown in our study. To clarify this point we have removed the reference to cytotoxicity as indicated below and can be found on line 206.

“We next investigated whether the increased anti tumor immunity observed following 5-OP-RU pulsing was a direct mechanism, or an indirect mechanism involving downstream effector cells.”

13) Importantly, no data are provided and no hypothesis is put forward to explain the negative regulation of NK cells by naïve MAIT cells (mouse or human).

Response: To interrogate this mechanism we have performed further experiments to investigate the phenotype of NK cells in B6-MAIT_{cast} MR1 WT and B6-MAIT_{cast} MR1^{-/-} mice. These experiments reveal that mature NK cell numbers are significantly enhanced in B6-MAIT_{cast} MR1^{-/-} mice compared to B6-MAIT_{cast} MR1 WT mice and this most likely accounts for the increased therapeutic effects observed when challenged with B16F10 tumor. Please refer to response to question 1 from Reviewer 2. The reasons for this difference are unclear and we agree this is worthy of further exploration, but without any clear candidates, this really requires a separate study of NK cell homeostasis and its regulation by MAIT cells, which we feel falls beyond the scope of this current study. This is reminiscent of similar phenotypes whereby mice lacking other innate cell lineages display increased numbers of other innate cell types (PMID 27668799).

General remark:

14) The authors state that B6-MAITCAST Mr1^{-/-} mice lack MAIT cells, but only provide 1 plot in Supplemental Figure 1A. Published data suggest that MAIT cells of the lung might only be mildly affected by the MR1-deficiency (Cui, JCI)

Response: This is not quite correct. The Cui *et al.* JCI 2015 study did not include the use of MR1 tetramers to identify MAIT cells, and therefore relied on a series of surrogate markers to try to identify these cells. One readout they relied on was the use of V α 19-J α 33 mRNA levels which reflects the invariant TCR- α chain used by MAIT cells. However, other T cells can use this TCR- α chain, which is why most T cells in V α 19J α 33 TCR transgenic C α knockout mice are not MAIT cells (PMID: 24101382). Recognising this, the authors of the Cui *et al.* JCI study suggested that MAIT cells likely fell within the CD44^{hi} ROR γ t/GFP⁺ DN/CD8^{lo} subset of cells, and V α 19J α 33 transcripts within these cells were diminished in MR1^{-/-} mice. The reviewer may be concerned by Figure 3C in that paper where they show that TCR- β ⁺Thy1.2⁺CD1d- α GC-Tet⁺GFP⁺CD4⁺ cells are not diminished in the absence of MR1, but the authors are not arguing that these are MAIT cells. They suggest that

MAIT cells fall within the DN and CD8^{lo} ROR γ t⁺ subsets which are diminished in the MR1^{-/-} mice in that study. Importantly, in a more recent paper from this group (PMID: 31431722) the authors use MR1 tetramers to show that MAIT cells are essentially absent in B6-MAIT_{cast} MR1^{-/-} mice and suggest that residual MR1 tetramer⁺ cells were naïve T cells that are functionally distinct from MAIT cells. And lastly, even if residual MAIT cells were present in B6-MAIT_{cast} MR1^{-/-} mice, MAIT cells by definition, are MR1-restricted and therefore cannot function in an antigen-responsive manner in the absence of MR1. Furthermore, responsiveness to 5-OP-RU antigen in the B6-MAIT_{cast} MR1^{-/-} was tested in **Figure 3E** and **Figure 4C**, and clearly shows that B6-MAIT_{cast} MR1^{-/-} mice do not respond to antigen.

15) The authors need to provide solid data to proof that in their hands B6-MAITCAST Mr1^{-/-} mice lack MAIT cells in the lung. This is an essential point for most experiments.

Response: The 5-OP-RU tetramer used in this study is recognized as the most definitive proof for the identification of MAIT cells and thus confirms that B6-MAIT_{cast} MR1^{-/-} mice lack MAIT cells. Please see our response to the previous point for an explanation of why there may be a misunderstanding of the Cui *et al.* JCI paper.

16) This MS contradicts an earlier report that activated MAIT cells would inhibit NK cells in a similar model of lung metastases. While the authors cite that work in the discussion and mention that the reason for the discrepancy is not clear, they fail to recognize that the published report is based on work in B6, while the current MS is based on work with congenic B6 that has increased numbers of MAIT.

Response: We are aware of this other report as noted in our discussion. As discussed above in response to point 1, we have reproduced the same data in C57BL/6 mice and this was present and clearly described in our original submission (**Figure 2D**, **Figure 2H**, **Figure 4G**, **Supplementary Figure 5A-B**). Therefore, the explanation proposed by the reviewer does not account for these differences.

17) Also, the designation of B6-MAITCAST mice as “MR1” and B6-MAITCAST MR1^{-/-} mice as “MR1^{-/-}” in the figures is misleading. Especially, as the authors often refer to the congenic B6-MAITCAST mice simply as “mice” in the text. As such it is not immediately clear if the data are derived from B6 or B6-MAITCAST mice. The authors need to clearly identify B6 and mice throughout the text and figures (not only figure legends)!

Response: We have clarified this as requested.

Specific comments

18) Figure 1A, B, E, F: comparison to wtB6 necessary

Response: Please refer to response to point 1 above.

19) How compare NK cells from B6-MAITCAST and mice B6-MAITCAST MR1^{-/-} mice functionally? Are the former less active: Killing of RMA-S, YAC-1, cytokine production?

Response: To address this question we have performed additional experiments comparing the cytotoxic activity of NK cells from B6-MAIT_{cast} and B6-MAIT_{cast} MR1^{-/-} mice. For these experiments we used YAC-1 and B16 F10 targets because experiments with B16F10 complement our *in vivo* data but require NK cells to be activated with IL-2 to elicit cytotoxic function against these targets. By contrast, YAC-1 cells can be killed by freshly isolated NK cells. With both tumor models we observed that NK cells derived from B6-MAIT_{cast} MR1 WT mice were not significantly less cytotoxic than those derived from B6-MAIT_{cast} MR1^{-/-} mice. This data is shown below as **Reviewer Only Figure 4** and supports our conclusions that the protected phenotype of B6-MAIT_{cast} MR1^{-/-} mice is predominantly associated with enhanced NK cell numbers.

Reviewer Only Figure 4: Cytotoxicity of NK cells from B6-MAIT_{cast} MR1 WT and B6-MAIT_{cast} MR1^{-/-} mice is comparable. **A.** Cells were cultured in 1000 IU/mL IL-2 for 5 days and then cocultured with Cr⁵¹-labelled B16F10 target cells. NK cell (Effector): tumor cell ratios (E:T ratios) are as indicated. Data is represented as the mean ± SEM of triplicate cultures from 5 pooled mice. **B.** Freshly isolated NK cells were cocultured with Cr⁵¹-labelled YAC-1 target cells. NK cell (Effector): tumor cell (Target) ratios (E:T ratios) are as indicated. Data is represented as the mean ± SEM of triplicate cultures from 3 combined experiments of n = 3 each.

18) What is the mechanism of negative regulation?

Response: Please refer to our response to point 13 above.

19) Figure 3 In vivo expanded MAIT cells inhibit tumor metastasis, have “normal” wtB6 in vivo expanded MAIT cells the same effect?

Response: Yes, this was shown in our original manuscript. Please refer to our response to point 1 above.

20) Too few info on E0771 tumors: growth in B6-MAITCAST Mr1^{-/-} not shown.

Response: We have already included data on in the E0771 model in the context of tumor cell-pulsing (**Figure 2B** and **Figure 2G**) and intranasal administration in both the prophylactic (**Figure 3F**) and therapeutic setting (**Figure 3G-H**) in the original manuscript. Given the large number of valuable suggestions for additional experimental studies suggested by the reviewers, we did our best to

prioritize what we thought would be the most important to achieve within a reasonable time frame to alleviate reviewer concerns. With this in mind, we felt that assessing growth of E0771 tumors in the lungs of B6-MAIT_{cast} MR1^{-/-} is a lower priority experiment that would not impact on the conclusion of our study.

21) Figure 4: Is 5-OP-RU specifically activating only MAIT cells?

Response: It is well accepted that 5-OP-RU specifically binds to the MAIT cell TCR and thus specifically activates MAIT cells (PMID 24695216). Our *in vivo* data indicates that the effects of either 5-OP-RU pulsing (**Figure 2E**) or intranasal delivery of 5-OP-RU antigen (**Figure 3E**) are attenuated in B6-MAIT_{cast} MR1^{-/-} mice (which lack MAIT cells) strongly suggesting that MAIT cells are required for this effect. To address this within our experimental system we have performed additional experiments that clearly show the effects of 5-OP-RU are attenuated when MAIT cells are not present. Please refer to point 6 from reviewer 2. Moreover, we provide new data (**Figure 4C**) showing that NK cells are not expanded in B6-MAIT_{cast} MR1^{-/-} mice and this data is referred to in the text on line 257:

“Notably, the increase in NK cells following 5-OP-RU intranasal administration was not observed in B6-MAIT_{cast} MR1^{-/-} mice, confirming the requirement of MAIT cells for this effect (**Figure 4C**).”

22) Figure 4E: B6-MAITCAST + 5-OP-RU vs untreated B6-MAITCAST Mr1^{-/-} = same outcome on number of metastases : how to explain? Also seen for wtB6 and wtB6 MR1^{-/-}?

Response: This data indicates that B6-MAIT_{cast} MR1^{-/-} mice are protected against B16F10 metastasis in the steady state but do not elicit an enhanced protection following intranasal delivery of 5-OP-RU (consistent with the lack of MAIT cells in these mice). This is the paradoxical finding that we describe in our paper including the abstract, where MAIT cells in resting, steady state, interfere with protective immunity, but when activated, they can promote tumor immunity.

23) The observed increase of MAIT cell genes revealed mainly MAIT17 (IL-17 producers) genes Rorc, Il17a, Cxcr6: not MAIT1 (IFN γ) specific genes though. Comments?

Response: Our data indicates that the production of IFN γ by MAIT cells peaks at day 2 and reduces to baseline by day 5 (**Figure 5C**). The genes presented in **Supplementary Table 1** (where lungs were harvested at day 5) are unlikely to reflect changes in MAIT cell activation, but rather reflect genes that are selectively expressed in MAIT cells as indicated in the text (lines 242). Notably our RNA-Seq data on purified MAIT cells (**Supplementary Figure 6A**) indicates that the expression of *Ifng* by MAIT cells is enhanced following 5-OP-RU treatment whereas *IL17a* expression was reduced.

24) Macrophages are found to be expanded and activated just like NK cells, but not followed up by the authors. What cytokines do the secreted macrophages produce? IFN γ ? IL-12, IL-15, IL-18?

Response: We believe that the activation of macrophages by MAIT cells is an interesting observation, and indeed we postulate on this in our discussion, but more extensive investigation of this falls beyond the scope of the current study. While it is unlikely that macrophages produce IFN γ as this is not generally considered to be a macrophage-derived cytokine, to satisfy the reviewer we have included data as **Reviewer Only Figure 5** that demonstrates that macrophages do not secrete IFN γ in response to 5-OP-RU stimulation. Given that we have not identified IL-12, IL-15 or IL-18 to be involved in

the mechanism, we believe that assessment of these cytokines was a lower priority experiment and so have not assessed these in our new data.

Macrophage IFN γ secretion

Reviewer Only Figure 5: *Macrophages do not secrete IFN γ in response to 5-OP-RU stimulation.* B6-MAIT_{cast} MR1 WT mice received 233 nM 5-OP-RU intranasally. Lungs were harvest a day later and production of IFN γ from macrophages (F4/80⁺CD64⁺) was assessed via flow cytometry. Data is presented as mean \pm SEM of n = 6- 7, Mann-Whitney test, ns = non-significant.

25) The increase of inflammatory response genes by NK cells from 5-OP-RU treated B6-MAITCAST mice supports the idea that inflammatory macrophages might activate NK cells.

Response: While this is possible, we do not see how this hypothesis can be derived from the inflammatory response gene signature of NK cells, when MAIT cells that produce cytokines, including IFN γ directly in response to 5-OP-RU, can activate NK cells and NK cell activation by 5-OP-RU is MAIT cell dependent.

26) Kinetic of IFN γ production by MAIT cells after treatment with 5-OP-RU: Day 2 is rather late effect:

Response: We believe that the reviewer may be inferring that because we have assessed cytokine production at day 2 that the effect of 5-OP-RU on MAIT cells could be indirect. However, 5-OP-RU is known to be a specific agonist for MAIT cells (PMID 24695216). Moreover, our new data with human cocultures clearly shows that MAIT cells are required for the effects mediated by 5-OP-RU (**Figure 6D**). Therefore, whilst it is likely that 5-OP-RU activates MAIT cells prior to the 2 day timepoint in our study, this does not affect the conclusions of our paper and so we have not addressed this in our study.

27) if gating on IFN γ ⁺ cells at this point, which cell types come up?

Response: Our data shows that MAIT cells are the predominant source of IFN γ (relative to conventional T cells) on a per cell basis (**Figure 5C**; **Supplementary Figure 4E**) and are required for the anti-tumor activity mediated by 5-OP-RU. However, we did not include a comprehensive lineage marker phenotypic analysis in these experiments and so are unable to answer this question

with our existing data. Although we did assess production of IFN γ by NK cells and showed that this was not increased following 5-OP-RU (**Reviewer Only Figure 6**; similarly to conventional T cells), we do not discount that other cell types may contribute to the production of IFN γ is outlined in our response to point 11 we have amended our discussion to reflect this.

Reviewer Only Figure 6: IFN γ secretion from NK cells is not increased by 5-OP-RU administration. B6-MAIT_{cast} MR1 WT mice received 233 nM 5-OP-RU intranasally. Lungs were harvested at day 2 and production of IFN γ from NK cells was assessed via flow cytometry. Data is presented as mean \pm SEM of n = 6-7.

28) Are NK cells already activated at day1 ?

Response: As referred to in our response to point 26, we believe the reviewer may be inferring that NK cells are directly activated by 5-OP-RU. Our new data where 5-OP-RU was added to NK cell cultures, clearly shows that NK cells are not directly activated by 5-OP-RU (**Figure 6D**). Furthermore, in the B6-MAIT_{cast} MR1^{-/-} mice which lack MAIT cells, NK cells are not expanded in response to 5-OP-RU, again confirming that NK cells do not directly respond to 5-OP-RU (**Figure 4C**). Therefore, we believe that assessment of NK cell activity at day 1 would not significantly strengthen our conclusions given that tumor challenge occurs at day 7 in this model.

29) Figure 5D: flaw in setup: IFN γ from NK cells is essential for response to B16; blocking IFN γ in this setting shows probably only that. Any impact of MAIT cell-derived IFN γ cannot be interpreted from this setup. Source of IFN γ not clear.

Response: As outlined above in response to points 11, 27 and 28, we acknowledge that the source of IFN γ cannot be definitively concluded from this *in vivo* experiment. We have added a sentence to the discussion to acknowledge this point on line 423.

30) Related: Supplemental Figure 6C: source of IFN γ not clear

Response: Whilst we clearly show that MAIT cells are required for 5-OP-RU mediated expansion of NK cells and produce high levels of IFN γ , we do not discount a potential role for other IFN γ -secreting cells in amplifying this effect. We have added text to acknowledge this as referred to above in response to point 29.

31) 5F: A2AR-/- adding this mutant is only distracting. Role of that molecule for NK cell anti-tumor response appear unrelated. Should be removed.

Response: We thank the reviewer for this comment, we agree that this may be distracting and have removed this data from the manuscript.

32) Figure 6: Human MAIT-NK cell cross talk results in increased activation of NK cells

Line: 308-310: statement is based on what? And why would that be good or important?

Response: Our data in **Supplementary Figure 7B** indicates that the genes comprising the gene list for the MAIT cell signature are very selectively expressed in MAIT cells. This is shown by the relatively high expression of these genes in MAIT cells but not other cell types. This is important because it validates this gene signature as a method to identify increased number of MAIT cells within a bulk tissue as is the case with the NK cell signature (See PMID 31088844). We have added additional text to clarify this point on line 330.

“Moreover, the ability of this gene signature to identify MAIT cells was comparable to the ability of a previously validated NK cell signature to identify NK cells (44) (**Supplementary Figure 7B**), thereby indicating that this gene signature can be used to evaluate the extent of MAIT cell infiltration in bulk tissues.”

33) Figure 6B: SKCM, BRCA TNBC: high MAIT score do not seem to reduce impact of a high NK cell score, but reduced the difference between impact of NK hi and NK lo scores by improving the impact of NK low scores! BLCA, CHOL: hi MAIT improves impact of both NK hi and lo scores

Response: This interpretation of the data is not valid since the graphs do not correspond to the same data between groups as assumed by the reviewer. For example for the SKCM group the NK high group is n = 238 (all patients, n = 81 MAIT high and n = 157 in MAIT low groups). This is because there is a correlation between NK and MAIT cell infiltrate meaning there are increased numbers of patients that are either “High” or “Low” for both subsets.

34) Figure 6C: purified NK cells +/- MAIT cells should be used

Response: We agree with the reviewer and have used purified NK cells and MAIT cells in **Figure 6D**. This data confirms that 5-OP-RU-mediated activation of MAIT cells enhanced NK cell activation. This new data further confirms the cross-talk relationship between activated MAIT cells and NK cells.

35) Figure 6E-G: legend in plots and diagrams misleading: according to the figure legend IL-2 is present in all settings

Response: We agree and have amended these figures as suggested.

36) Figure 6G actually indicates that 5-OP-RU has an effect on NK cells as it alone (on IL-2 preactivated cells) can induce IFN γ production from around 60% of NK cells – just like IL-2. But what is missing here is the control, i.e., PBMC prepared as before but cultured without IL-2 for the last 4hrs.

Response: We believe that the reviewer has misinterpreted this data based upon the above point (#35) as these values were indicative of baseline (IL-2 only) NK cell activation. We apologize for this confusion. In addition, we have now clearly provided evidence that 5-OP-RU cannot directly stimulate NK cells (**Figure 6D**).

37) Along the same line as before, a MAIT-NK cross talk cannot be deduced from the experimental data as PBMCs and not purified populations were used.

Response: Please see above response, we have now included experiments with enriched MAIT cell and NK cell populations that address this

38) Minor: 178: onto not into, Line 206: 333WT mice? Should be B6-MAITCAST mice

Response: We thank the reviewer for noticing these two minor errors in the manuscript and have since amended them.

Reviewer #2 (Innate immunity, innate lymphoid cells, MAIT cell) (Remarks to the Author):

The paper is potentially interesting but, at this stage, some of the data is confusing. While the different MAIT cell-dependent regulation of NK cells in the two settings is novel and interesting, the mechanism for this should be clarified. Thus, the main concept of this paper should be better demonstrated. Two opposite effects of MAIT cells in tumors are described. In both cases MAIT cell impact is dependent on NK cell regulation but drives contrasting outcomes in the modulation of tumor immunosurveillance. The divergent role of MAIT cells in tumors (in particular mucosal tumors) has already been described, showing that MAIT cells can be pathogenic producing IL-17 and IL-22 or protective having a type 1 signature. How are MAIT cells differently affecting NK cell effector functions in the two experimental settings?

Response: We thank the reviewer for the positive appraisal of our manuscript. We believe that our responses to the specific comments below will clarify all of the aspects that the reviewer has raised here, particularly with regard to how MAIT cells affect NK cell effector function in the two experimental settings. Furthermore, we trust that the reviewer will find that the new data enhances the interesting nature of our findings.

Specific comments are listed below.

1) MR1-deficient mice are protected from tumor growth in the B16 model and this is dependent on NK cell activation (Figure 1). A more careful analysis of NK cell activation (cytokine production, cytotoxicity markers) would be helpful. How are MAIT cells negatively impacting on NK cell activation in this context?

Response: To address this question we have performed additional experiments to comprehensively analyze both the numbers of NK cells and expression of NK cell activation markers on NK cells isolated from the lungs of B6-MAIT_{cast} MR1 WT and B6-MAIT_{cast} MR1^{-/-} tumor-bearing mice. NK cell phenotype was assessed at day 5 post B16F10 tumor challenge.

These experiments revealed that in B6-MAIT_{cast} MR1^{-/-}, but not B6-MAIT_{cast} MR1 WT mice, injection of B16F10 cells led to a significant increase in NK cells within the lungs of tumor bearing mice. However, expression of the NK cell receptors DNAM-1, CD96 and NKG2D, as well as the cytokines IFN γ and TNF, and the degranulation marker CD107a were similar in NK cells derived from tumor bearing B6-MAIT_{cast} MR1 WT and B6-MAIT_{cast} MR1^{-/-} mice. This new data supports our hypothesis that the enhanced killing of B16F10 lung metastasis in the B6-MAIT_{cast} MR1^{-/-} mice is due to the increased number of NK cells, and not an increase in function or cytotoxic potential on a per-cell basis. This new data is shown in **Supplementary Figure 1G-H** and referred to on line 166 of the revised manuscript as stated below:

“To investigate the mechanism by which B6-MAIT_{cast} MR1^{-/-} mice were protected against B16F10 lung metastasis, we investigated the phenotype of NK cells in tumor-bearing mice at day 5 post-tumor challenge. This analysis revealed that whilst NK cell numbers were significantly increased in the lungs of B6-MAIT_{cast} MR1^{-/-} mice following tumor challenge, this effect was not observed in B6-MAIT_{cast} MR1 WT mice (**Supplementary Figure 1G**). Expression of the NK cell receptors DNAM-1, CD96 and NKG2D, as well as the cytokines IFN γ and TNF, and the degranulation marker CD107a

were similar in NK cells derived from tumor bearing B6-MAIT_{cast} MR1 WT and B6-MAIT_{cast} MR1^{-/-} mice (**Supplementary Figure 1H**). These results suggest that the enhanced anti-metastatic response observed in B6-MAIT_{cast} MR1^{-/-} mice is due to increased NK cell numbers.”

2) Antigen-activated MAIT cells are beneficial in tumor models and this is still dependent on NK cells but having the opposite effect (Figure 2). How do the authors explain that no difference was observed in the presence or absence of MAIT cells using tumors pulsed with a MAIT specific antigen (Figure 2C)? It looks like the effect is not dependent on MAIT cells.

Response: 5-OP-RU pulsing of tumor cells results in a reduction in metastasis in B6-MAIT_{cast} MR1 WT mice. However, this effect was not observed in B6-MAIT_{cast} MR1^{-/-} mice that lack MAIT cells. Therefore, there is in fact a clear difference in the presence or absence of MAIT cells, thus confirming that the anti-tumor effects are indeed MAIT cell dependent. We apologize if this was not clearly explained and to rectify this, we have altered the text accordingly. Furthermore, we have revised **Figure 2C-E** to include additional graphs, which indicates the fold change in number of metastases in each strain. This new data clearly shows a significant reduction in metastasis in MAIT cell replete mice (**Figure 2C-D**) but not MAIT cell deficient mice (**Figure 2E**). This is referred to on line 182 of the revised manuscript:

“Furthermore, the number and fold-change of 5-OP-RU-pulsed tumor cell metastases were much lower than non-pulsed B16F10 tumors in B6-MAIT_{cast} MR1 WT mice (**Figure 2C**) and C57BL/6 WT mice (**Figure 2D**). However, no reduction in tumor metastases following 5-OP-RU pulsing was observed in B6-MAIT_{cast} MR1^{-/-} mice (**Figure 2E**), confirming the absolute requirement for MAIT cells in this protective effect.”

3) The authors show that MR1 upregulation per se does not impact the phenotype observed (Supplementary Figure 2). They show no effect using the E0771 cell line where they say that MR1 is not upregulated. MR1 is actually absent in this case (Figure 2B). Maybe the antigen pulsing did not work or MR1 is not expressed at all in this cell line. I think this experiment should be revised and more clearly explained.

Response: This series of experiments indicate that in the B16F10 model, tumor cells present 5-OP-RU upon antigen pulsing and this subsequently reduces their metastatic potential. By contrast in the E0771 model, 5-OP-RU pulsing results in significantly less MR1 upregulation, which likely explains the lack of effect of 5-OP-RU on metastasis in this pulsing model. To confirm this, we repeated the experiments measuring MR1 expression following 5-OP-RU pulsing on B16F10 and E0771 cells in parallel. This confirmed that MR1 is upregulated on B16F10 cells to a much greater extent than on E0771 cells. This new data is shown in revised **Figure 2B**.

4) In the Supplementary figure 2, the authors also show that tumor growth is not impacted in the absence of antigen pulsing comparing B16cherry cells and B16MR1-overexpressing cells (last panels). Actually, with the antigen there is no difference between the two cell lines either (first panels). Why are the authors now using B16 cherry cells?

Response: We apologize if this was unclear in the original submission. The B16F10cherry cells are the relevant control for the B16F10-MR1-overexpressing cells since the cells were transduced with either an MSCV-Cherry or MSCV-MR1-Cherry retrovirus. This is now more clearly explained in the results section (see below).

With regard to the data, we agree that MR1-overexpressing tumors do not exhibit an enhanced response in this regard, and thus our conclusion is in agreement with the reviewer. We hypothesize that this indicates that the basal level of MR1 expression is sufficient to elicit maximal therapeutic effects under these experimental conditions. To investigate this further, we have performed additional experiments to generate MR1-deficient B16F10 tumor cells using CRISPR/Cas9 technology. These experiments reveal that expression of MR1 on B16F10 tumor cells was required for therapeutic activity in this pulsing model, consistent with our hypothesis. This data is shown in **Supplementary Figure 2D** and **Figure 2F** and is referred to on line 188.

“In these experiments, we also utilized a MSCV-cherry retroviral vector to overexpress MR1 on B16F10 cells (B16F10cherryMR1) and compared tumor growth to control B16F10 cells (B16F10cherry). MR1 overexpression resulted in an increase in cell surface MR1 expression, but did not significantly affect the extent of B16F10 metastases either in the presence or absence of antigen pulsing (**Supplementary Figure 2A-C**). This suggests that endogenous MR1 expression on B16F10 tumor cells is sufficient for the anti-tumor response observed following pulsing with MAIT cell antigens. To confirm this, we evaluated the effect of 5-OP-RU pulsing on the metastasis of B16F10 tumor cells following CRISPR/Cas9-mediated deletion of MR1. Having verified successful knockout of MR1 on B16F10 tumor cells (**Supplementary Figure 2D**), we observed that 5-OP-RU no longer exhibited an anti-tumor effect (**Figure 2F**), thus confirming the requirement for tumor-derived MR1 expression in this antigen-pulsing model.”

5) Intranasal exposure to a specific MAIT antigen is sufficient to promote tumor control (Figure 3). How do the authors explain that the E0771 model works in this case? If the tumors still do not express MR1 it means that MAIT cell activation fully occurred in the lung and it is sufficient to provide protection in the tumor.

Response: The data in Figure 3 highlights the therapeutic activity of 5-OP-RU when administered via intranasal delivery. In this setting MAIT cells can be activated via MR1 expressed on non-tumor cells. We have previously shown that intranasal administration of 5-OP-RU expands MAIT cells in non-tumor bearing mice (PMID 27143301) consistent with MR1’s role as a major histocompatibility complex-related protein and its expression on a range of immune cells. In the present study, 5-OP-RU was administered on days 0, 1, 2 and 4 prior to tumor cell challenge on day 7. MAIT cell expansion was observed at day 6 (prior to tumor challenge). Our data enhances the therapeutic relevance of MAIT cell antigen stimulation given that these effects are not dependent on the levels of MR1 expressed on the tumor cells.

6) NK cell activation is the proposed mechanism underlying the reduced tumor growth (metastases) upon antigen-driven activation of MAIT cells in both the mouse model (Figures 4-5) and in vitro experiments using human cells (Figure 6). How is this occurring?

Response: Our data indicates that the anti-tumor effect observed following 5-OP-RU mediated activation of MAIT cells requires NK cells (**Figure 4F-G**). Analysis of the phenotype of NK cells by RNA-seq revealed that an IFN γ response was one of the most highly upregulated pathways following MAIT cell activation (**Figure 5A-B**). Subsequently, MAIT cells were confirmed to be major producers of IFN γ following intranasal delivery of 5-OP-RU (**Figure 5C**) and IFN γ was demonstrated to be required for both the expansion of NK cells (**Supplementary Figure 6C**) and the therapeutic response (**Figure 5D**). Our data thus supports a model whereby MAIT cell-derived IFN γ activates NK cells leading to enhanced anti-tumor activity.

7) In the in vitro experiments (Figure 6), is the effect of 5-OP-RU dependent on MAIT cells?

Response: Yes. To more clearly address this question we have performed additional experiments. Human NK cells were isolated and cultured alone or with MAIT cells in the presence or absence of 5-OP-RU antigen. The activation of NK cells was assessed by IFN γ production at 16 hours post activation. This new data shows that 5-OP-RU antigen stimulation has no effect on NK cell function in the absence of MAIT cells and further supports our data in the mouse models that antigen-specific stimulation of MAIT cells results in NK cell activation and robust anti-tumor responses. This is referred to in **Figure 6D** and the following text on line 353.

“To confirm that this was a MAIT cell-mediated effect, we isolated NK cells and MAIT cells from healthy donors and cultured these cells alone or together in IL-2 in the presence or absence of 5-OP-RU. Stimulation of NK cells alone with 5-OP-RU did not result in increased production of IFN γ . However, stimulation of NK cells cultured with MAIT cells and 5-OP-RU resulted in significantly increased production of IFN γ by NK cells (**Figure 6D**), confirming that 5-OP-RU-mediated stimulation of MAIT cells subsequently induces potent activation of NK cells.”

8) The analysis of the MAIT cell gene signature and the correlation with clinical data in TCGA cohorts is interesting and clear but the methods should be clarified.

Response: We have clarified this by extending the methods section with regard to this analysis. These revised methods are located from line 560.

“MAIT and NK signature generation

To calculate a gene signature associated with MAIT cells, RNA-Seq profiling of 29 different PBMCs were downloaded from the Gene Expression Omnibus (Accession ID GSE107011). Differential expression was performed using limma-voom between the MAIT cell group and each other cell type. A MAIT cell gene set was calculated by selecting the genes that had nominal p value < 0.05 and log-fold-change > 0 against all other cell types. For NK cells, an established gene set was used (44). KLRB1 was excluded from the signatures due to being in both original gene sets. Signature scores were calculated using the singscore R package (61). Heatmaps for GSE107011 were generated from gene-wise Z-scores of log-counts-per-million using the ComplexHeatmap R package (62).

Survival Analysis

Samples were classified into "High" and "Low" categories for MAIT and NK cell groups based on whether a sample had a signature score above or below (respectively) the mean value of that cell type's signature scores within that TCGA cohort. Progression-free survival (PFS) endpoints were matched to samples from an integrated TCGA pan-cancer clinical data resource (63). Kaplan-Meier and Cox regression models on different subsets of the TCGA data using the "survival" R package (version 3.1-8)."

9) The analysis between multiple groups should be performed and a proper statistical test (Anova) should be used throughout the paper (Figure 1F, Figure 2C, Supplementary Figure 2...).

Response: We have revised the statistical analysis to ANOVA where appropriate including the data for **Figure 1F**. For **Figure 2C** and **Supplementary Figure 2**, a Mann-Whitney Test was performed since there are only two groups. Importantly, the use of a Mann-Whitney test is more appropriate for these data because a student's t test assumes normal distribution of data points, and equal variance between groups, and these are not assumptions that we can make with confidence. However, we would add for the reviewer's information that the student's t test also showed that these data were statistically significant (**Figure 2C** $**p < 0.001$; **Supplementary Figure 2A** $**p < 0.001$ (left), $*p < 0.05$ (right)).

Minor points:

1) Figures layout and panel numbering should be improved. Figures letters are missing and it was very complicated to follow the text.

Response: We have amended figure layouts as requested.

2) Is the quantification of MAIT and NK cell activation done in tumor bearing mice? A proper quantification of the markers and statistical analysis should be performed. (Supplementary Figure 1).

Response: We have performed additional experiments to address this point. Please refer to our response to point 1 above.

3) B16 s.c. experiment (Supplementary Figure 1) in MR1-deficient mice with or without NK cells depletion would be informative.

Response: Whilst we agree this could be informative, we felt that these experiments were probably lower priority than the others that were prompted by this and other reviewers, and we believe they were not necessary for the interpretation and conclusions in this paper.

REVIEWER COMMENTS

Reviewer #1 (Remarks to the Author):

The authors addressed most points raised sufficiently.

Reviewer #2 (Remarks to the Author):

The authors have thoroughly addressed my critiques

Reviewer #3 (Remarks to the Author):

The study is much improved but a significant issue revolves around the sole use of anti-ASGM1 as means to demonstrate NK-specific effects. ASGM1 has been well documented to be expressed on macrophages and activated T cells and simply cannot be used as a means to demonstrate NK-specific effects. Anti-NK1.1 is considered to be more NK specific (although also on NK/T cells) and should be used instead. This is particularly important given the opposing effects observed with the KO mice versus using MAIT activation models.

Reviewer #1 (Remarks to the Author):

The authors addressed most points raised sufficiently.

Reviewer #2 (Remarks to the Author):

The authors have thoroughly addressed my critiques

Response: We are pleased that reviewer 1 and 2 are satisfied with the additional changes to the manuscript.

Reviewer #3 (Remarks to the Author):

1) The study is much improved but a significant issue revolves around the sole use of anti-ASGM1 as means to demonstrate NK-specific effects. ASGM1 has been well documented to be expressed on macrophages and activated T cells and simply cannot be used as a means to demonstrate NK-specific effects. Anti-NK1.1 is considered to be more NK specific (although also on NK/T cells) and should be used instead. This is particularly important given the opposing effects observed with the KO mice versus using MAIT activation models.

Response: We thank reviewer 3 for reviewing our changes to the manuscript in response to reviewer 1 and 2, and are pleased that they agree the manuscript is much improved. To address the additional question raised by reviewer 3 on using anti-NK1.1 to target NK cells, we have performed further experiments to determine the effect of this antibody on B16F10 lung metastasis in B6-MAIT_{cast} MR1 WT versus B6-MAIT_{cast} MR1^{-/-} mice and in the model where MAIT cells are activated following 5-OP-RU intranasal administration.

These new experiments revealed that the enhanced anti-tumour immune response observed in B6-MAIT_{cast} MR1^{-/-} mice compared to B6-MAIT_{cast} MR1 WT mice was attenuated when NK cells were targeted with either anti-ASGM1 (**Figure 1F**) or anti-NK1.1 (**Supplementary Figure 1F**). Notably, these experiments were conducted in parallel and there was no difference in tumour burden observed following treatment with either of the antibodies.

Our original data in Figure 4 shows that the reduction in B16F10 metastasis following 5OP-RU-mediated MAIT cell activation was NK cell-dependent. This was confirmed by NK cell depletion with anti-ASGM1 (**Figure 4F**) or in NKp46^{cre}Mcl-1^{loxP} mice, which lack NK cells (**Figure 4G**). To further support this data, we performed a new experiment using the anti-NK1.1 antibody in the context of MAIT cell expansion (**Supplementary Figure 5A**). This new data clearly confirms that the MAIT cell-mediated anti-tumour effect is lost when NK cells are targeted with anti-NK1.1, to a similar extent as anti-ASGM1. Taken together, this data clearly shows that activated MAIT cells require NK cells for their anti-tumor efficacy. Thus this new data unequivocally supports our conclusion that MAIT cells regulate NK cell mediated tumor immunity in both the naïve and activation settings.

REVIEWERS' COMMENTS

Reviewer #3 (Remarks to the Author):

The new data with anti-NK1.1 and the KO mice have greatly strengthened the study and conclusions that NK cells are indeed responsible. The authors are commended for doing the studies. All issues addressed.